# BLEND: A Benchmark for LLMs on Everyday Knowledge in Diverse Cultures and Languages

**Junho Myung**[1,*], **Nayeon Lee**[1,*], **Yi Zhou**[2,*], **Jiho Jin**[1], **Rifki Afina Putri**[1],
**Dimosthenis Antypas**[2], **Hsuvas Borkakoty**[2], **Eunsu Kim**[1], **Carla Perez-Almendros**[2],
**Abinew Ali Ayele**[3,4], **Víctor Gutiérrez-Basulto**[2], **Yazmín Ibáñez-García**[2], **Hwaran Lee**[5],
**Shamsuddeen Hassan Muhammad**[6], **Kiwoong Park**[1], **Anar Sabuhi Rzayev**[1], **Nina White**[2],
**Seid Muhie Yimam**[3], **Mohammad Taher Pilehvar**[2], **Nedjma Ousidhoum**[2],
**Jose Camacho-Collados**[2], **Alice Oh**[1]

[1]KAIST, [2]Cardiff University, [3]Universität Hamburg, [4]Bahir Dar University,
[5]NAVER AI Lab, [6]Imperial College London

## Abstract

Large language models (LLMs) often lack culture-specific knowledge of daily life, especially across diverse regions and non-English languages. Existing benchmarks for evaluating LLMs' cultural sensitivities are limited to a single language or collected from online sources such as Wikipedia, which do not reflect the mundane everyday lifestyles of diverse regions. That is, information about the food people eat for their birthday celebrations, spices they typically use, musical instruments youngsters play, or the sports they practice in school is common cultural knowledge but uncommon in easily collected online sources, especially for underrepresented cultures. To address this issue, we introduce **BLEND**, a hand-crafted benchmark designed to evaluate LLMs' everyday knowledge across diverse cultures and languages. BLEND comprises 52.6k question-answer pairs from 16 countries/regions, in 13 different languages, including low-resource ones such as Amharic, Assamese, Azerbaijani, Hausa, and Sundanese. We construct the benchmark to include two formats of questions: short-answer and multiple-choice. We show that LLMs perform better for cultures that are highly represented online, with a maximum 57.34% difference in GPT-4, the best-performing model, in the short-answer format. For cultures represented by mid-to-high-resource languages, LLMs perform better in their local languages, but for cultures represented by low-resource languages, LLMs perform better in English than the local languages. We make our dataset publicly available at: `https://github.com/nlee0212/BLEnD`.

## 1 Introduction

Despite the worldwide usage of large language models (LLMs), capturing cultural everyday knowledge specific to a particular country or region is challenging because such knowledge is often not explicitly documented in online data sources like Wikipedia, which are commonly used to train LLMs. For instance, the answers to mundane everyday questions such as *"What can typically be found in the backyard of houses in your country?"* are not included in the training data of LLMs, except for a handful of highly represented regions such as North America. Consequently, LLMs may provide incorrect, incomplete, or nonsensical responses to everyday questions in underrepresented cultures,

---

*Equal contribution.

*Co-first authors: junho00211@kaist.ac.kr, nlee0212@kaist.ac.kr, zhouy131@cardiff.ac.UK

38th Conference on Neural Information Processing Systems (NeurIPS 2024) Track on Datasets and Benchmarks.

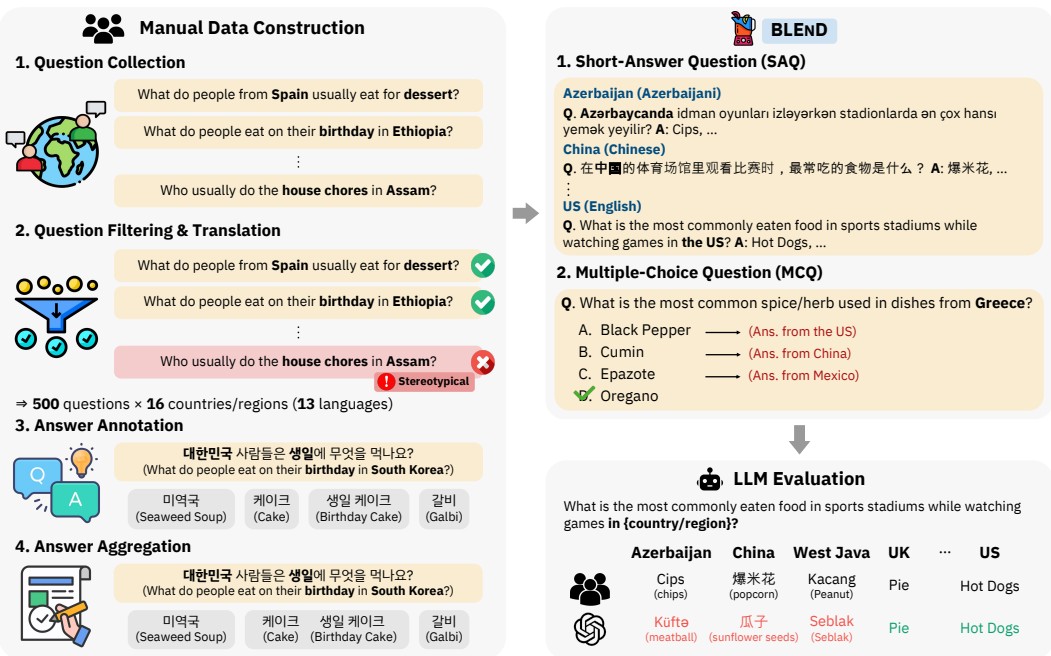

Figure 1: The overall framework of dataset construction and LLM evaluation on BLEND. BLEND is built through 4 steps: question collection, question filtering & translation, answer annotation, and answer aggregation. The dataset includes the same questions in 13 different languages, answered from 16 different countries/regions. We evaluate LLMs by short-answer and multiple-choice questions.

even though these inquiries are frequently encountered in daily lives. This can lead to hallucinations or stereotypical responses, potentially offending a large and diverse user base.

This challenge becomes even more evident in cross-lingual settings, as most LLMs are primarily trained on English data reflecting Western perspectives [8, 20, 15]. They often reflect the stereotypes present in the training data [19, 18, 21, 36, 13], hence these models would often respond based on Western perspectives rather than reflecting actual diverse practices. Ideally, language models would reflect the cultural norms of various regions around the world and generate culturally appropriate content when responding in local languages of the regions, unless otherwise specified. To develop multilingual LLMs with such cultural appropriateness, we first need to evaluate the cultural commonsense knowledge. However, there is no well-crafted multilingual multicultural benchmark that captures the daily lives of people in diverse cultures.

To bridge this gap, we present **BLEND**, a **B**enchmark for **L**LMs on **E**veryday k**n**owledge in **D**iverse cultures and languages. The benchmark covers 13 languages spoken in 16 different countries and regions shown in Table 1. Note that we include languages that are spoken in two regions with vastly different cultures, such as South Korea and North Korea, both represented by the Korean language. To effectively capture the cultural diversity of people's daily lives, we recruit annotators who are native speakers from various countries. The final dataset includes 500 socio-cultural question-answer pairs for each country/region in 6 categories: *food*, *sports*, *family*, *education*, *holidays/celebrations/leisure*, and *work-life*. To capture a comprehensive understanding of the cultural sensitivity of LLMs, we create a set of questions and answers in two formats: short-answer and multiple-choice questions. The overall framework for construction and evaluation of BLEND is shown in Figure 1. The statistics of BLEND are shown in Table 1 [1]. In total, BLEND features an extensive collection of 52.6k question-and-answer pairs, 15k short-answer and 37.6k multiple-choice.

Our experimental results on BLEND show that even current state-of-the-art LLMs exhibit unbalanced cultural knowledge and unfair cultural biases across various countries and regions. The average performance of all tested models on short answer questions about United States (US) culture in English is 79.22%. In contrast, when asked about Ethiopian (ET) culture in Amharic, the average performance

---

[1]Throughout the paper, we use the two-letter ISO codes for each country/region and language, as shown in Table 3.

Table 1: Statistics of the question samples within BLEND. BLEND is composed of two question types: Short Answer Questions (SAQ) and Multiple-Choice Questions (MCQ). The question samples are generated based on the 500 question templates generated by annotators from all countries/regions.

| | | SAQ | | MCQ | |
|---|---|---|---|---|---|
| **Country/Region** | **Language** | **Count** | **Language** | **Count** |
| United States (US) | English (en) | 500 | | 1,942 |
| United Kingdom (GB) | English (en) | 500 | | 2,167 |
| China (CN) | English (en), Chinese (zh) | 1,000 | | 1,929 |
| Spain (ES) | English (en), Spanish (es) | 1,000 | | 1,931 |
| Indonesia (ID) | English (en), Indonesian (id) | 1,000 | | 1,995 |
| Mexico (MX) | English (en), Spanish (es) | 1,000 | | 1,899 |
| South Korea (KR) | English (en), Korean (ko) | 1,000 | | 2,512 |
| Greece (GR) | English (en), Greek (el) | 1,000 | English (en) | 2,734 |
| Iran (IR) | English (en), Persian (fa) | 1,000 | | 3,699 |
| Algeria (DZ) | English (en), Arabic (ar) | 1,000 | | 2,600 |
| Azerbaijan (AZ) | English (en), Azerbaijani (az) | 1,000 | | 2,297 |
| North Korea (KP) | English (en), Korean (ko) | 1,000 | | 2,185 |
| West Java (JB) | English (en), Sundanese (su) | 1,000 | | 2,345 |
| Assam (AS) | English (en), Assamese (as) | 1,000 | | 2,451 |
| Northern Nigeria (NG) | English (en), Hausa (ha) | 1,000 | | 2,008 |
| Ethiopia (ET) | English (en), Amharic (am) | 1,000 | | 2,863 |
| **Subtotal** | | 15,000 | | 37,557 |
| **Total** | | | | 52,557 |

drops to only 12.18%, highlighting a significant performance gap in relatively underrepresented cultures and languages. A similar trend is observed in the multiple-choice format, where the LLMs are required to choose the correct answer for each target country/region, with answers from other countries/regions presented as wrong options.

The main contributions of our paper are as follows:

- We present BLEND, a benchmark of carefully crafted 52.5k question-answer pairs that reflect the everyday cultural knowledge across 16 countries/regions in 13 different languages.

- Within BLEND, we propose two types of questions to automatically measure the cultural knowledge in LLMs: short-answer questions and multiple-choice questions.

- We conduct extensive experiments across 16 LLMs on BLEND, showing a significant performance gap between highly represented cultures and underrepresented cultures.

## 2 Related Work

Although LLMs generally incorporate extensive parametric knowledge from large text corpora during pretraining [25], such models frequently display bias due to imbalanced representations in the data sources [3]. Cultural knowledge is critical in enhancing the reasoning capabilities of LLMs, contributing significantly to their success across various downstream applications.

Numerous studies have examined the socio-cultural aspects of LLMs. Previous work on cultural NLP defines culture as the way of life of a specific group of people [10]. Most research on the cultural knowledge of LLMs centers on the culture at a national level. Anacleto et al. [1] collect commonsense knowledge about eating habits in Brazil, Mexico, and US through the Open Mind Common Sense portal. GeoMLAMA [33] introduces 16 geo-diverse commonsense concepts and uses crowdsourcing to compile knowledge from 5 different countries, each in its native languages. Nguyen et al. [22] introduce a methodology to extract large-scale cultural commonsense knowledge from the Common Crawl corpus on geography, religion, and occupations. CREHate [17] is a cross-cultural English hate speech dataset covering annotations from 5 English-speaking countries. CultureAtlas [9] includes textual data encapsulating the cultural norms from 193 countries, primarily sourced from Wikipedia documents in English. However, the majority of these studies are conducted exclusively in English and focus on more objective aspects of culture that are written in formal data sources.

Table 2: Detailed statistics of the number of questions per category for each country/region in Short Answer Questions (SAQ) and Multiple-Choice Questions (MCQ).

|  | **Food** | **Sports** | **Family** | **Education** | **Holidays** | **Work-life** |
|---|---|---|---|---|---|---|
| **SAQ** | 105 | 88 | 63 | 84 | 92 | 68 |
| **MCQ** | | | | | | |
| United States (US) | 642 | 393 | 60 | 173 | 500 | 174 |
| United Kingdom (GB) | 990 | 403 | 50 | 189 | 427 | 108 |
| Spain (ES) | 714 | 476 | 43 | 172 | 425 | 101 |
| Mexico (MX) | 489 | 491 | 39 | 183 | 578 | 119 |
| Indonesia (ID) | 471 | 369 | 60 | 212 | 699 | 184 |
| China (CN) | 475 | 349 | 74 | 200 | 705 | 126 |
| South Korea (KR) | 753 | 792 | 57 | 218 | 539 | 153 |
| Algeria (DZ) | 873 | 569 | 59 | 189 | 819 | 91 |
| Greece (GR) | 1,345 | 516 | 40 | 154 | 500 | 179 |
| Iran (IR) | 666 | 519 | 50 | 173 | 2,135 | 156 |
| North Korea (KP) | 784 | 430 | 78 | 228 | 476 | 189 |
| Azerbaijan (AZ) | 852 | 513 | 65 | 216 | 453 | 198 |
| West Java (JB) | 892 | 461 | 20 | 160 | 680 | 132 |
| Assam (AS) | 862 | 584 | 34 | 198 | 666 | 107 |
| Northern Nigeria (NG) | 647 | 421 | 50 | 207 | 508 | 175 |
| Ethiopia (ET) | 984 | 649 | 46 | 278 | 692 | 214 |

More recent studies have focused on the cultural knowledge of non-English speaking countries and languages. For instance, CLIcK [14] and HAE-RAE Bench [29] evaluate LLMs' knowledge in Korean, while COPAL-ID [32], ID-CSQA [26], and IndoCulture [15] include culturally nuanced questions in Indonesian. Nonetheless, we do not know of any work that has been done to compare the cultural adaptiveness of LLMs across diverse languages and cultures using the same question set, which would enable a direct comparison.

Other recent work focuses on capturing the everyday cultural nuances of LLMs using social networking platforms. StereoKG [7] extracts cultural stereotypes of five nationalities and five religious groups from questions posted on X (formerly Twitter) and Reddit. However, this method produces a significant amount of noisy and inappropriate assertions due to insufficient filtering. CAMeL [20] includes masked prompts from naturally occurring contexts on X, focusing on Arabic content, and CultureBank [28] is a collection of diverse perspectives and opinions on cultural descriptors, including English comments from TikTok and Reddit. However, these datasets are limited to a single language and rely solely on data available from social media, not able to capture people's everyday behaviors to the full extent [31].

In contrast to prior work, BLEND is carefully human-crafted, capturing everyday life cultural knowledge across 13 languages spoken in 16 different countries/regions including underrepresented regions such as West Java and North Korea.

## 3 Construction of BLEND

**Language Coverage.** We select languages with varying levels of resource availability using the metrics defined by Joshi et al. [12]. The resource availability of languages included in BLEND is shown in Table 4 in the Appendix. Additionally, we involve at least one author who is a native speaker of the language and originally from the country/region represented in the dataset to handle the data inspection process[2].

**Question Collection and Filtering.** BLEND includes 500 question templates that reflect daily life aspects across six socio-cultural categories: *food*, *sports*, *family*, *education*, *holidays/celebrations/leisure*, and *work-life*. To create these templates, we collect 10-15 questions for each category from at least two native annotators per country/region. These annotators are asked to generate culturally relevant questions about their countries while avoiding stereotypical questions. The question generation guideline is shown in Appendix B.4. The collected questions are filtered

---

[2]North Korea was an exception, where we collaborated with a South Korean researcher studying North Korean language.

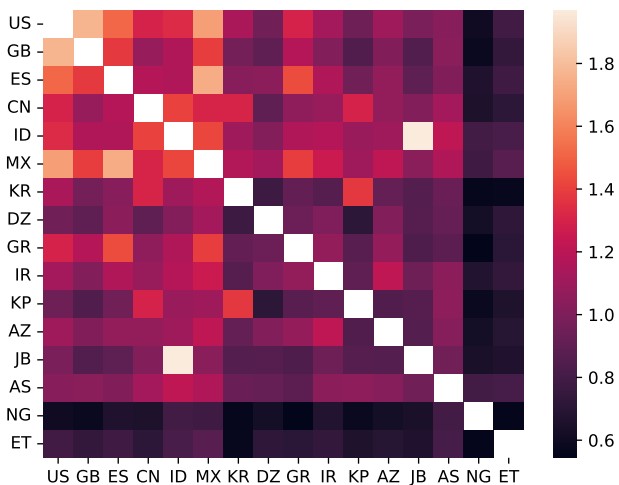

Figure 2: Heatmap showing the average number of common lemmas within each question between all country/region pairs. Pairs from the same countries/regions are shown in white. Higher numbers of shared lemmas indicate that those countries/regions provide more similar answers compared to other countries/regions (e.g., Indonesia and West Java).

to eliminate duplicates and country-specific items that can only apply to one country/region. For example, items with proper nouns from a single country/region are excluded. Then the questions are formatted into templates like "*What is a common snack for preschool kids in **your country**?*" Subsequently, '***your country***' is replaced by the country/region names for localizing the questions. Except for US and GB, the questions are translated into the local languages by the native speakers. This process results in a comprehensive dataset of 15,000 short-answer questions, as shown in Table 1. The specific number of questions per topic is shown in Table 2.

**Answer Annotation.** To obtain the answers to the collected questions, we recruit annotators who are native speakers of the target languages and are originally from the target regions/countries. We ensure that the annotators have lived in these countries for over half of their lifetimes [3]. For most countries, we recruit annotators through Prolific [4]. However, in cases where it is not possible to find annotators through crowdsourcing platforms (i.e., DZ, KR, KP, AZ, JB, AS, NG, and ET), we directly recruit five annotators who meet our criteria [5].

Annotators are required to give at least one short answer to each question and can offer up to three responses if a single answer is insufficient. If an annotator does not know the answer, they can choose from the following options: *'not applicable to our culture,'* *'no specific answer for this question,'* *'I don't know the answer,'* or *'others.'* By default, responses are collected from five annotators per question. If an annotator chooses *'I don't know the answer'*, we discard the response and collect a new one. This process continues until five valid responses for each question are obtained, or more than five annotators choose *'I don't know'*. Examples of the collected questions with answers from each country are presented in Figure 1. The guideline and the interface for answer annotation provided to annotators are shown in Appendix B.5 and B.6.

**Answer Aggregation.** We request 1-2 annotators from each country to review the annotations and remove invalid answers. These invalid answers appear to be due to some annotators misunderstanding a question, leading to nonsensical answers. Additionally, due to the nature of natural language, there are multiple variations of a single term (e.g., "go to bed" and "sleep"). We instruct the annotators to group these variants into one to ensure the final dataset contains accurate vote counts for each answer. We also ask the annotators to translate all the annotations into English. As a result, our final dataset includes variants in local languages and English, along with a final vote count for answers to the question.

---

[3]This condition was not fully met for North Korea due to a very limited pool of annotators.

[4]https://www.prolific.co/

[5]Tables 5 and 6 in the Appendix shows a detailed demographic distribution of the annotators.

**Statistical Analysis on Annotations.** We analyze the annotations to assess their quality and consistency, as detailed in Table 7 in the Appendix. Despite the subjective nature of the questions, the average level of agreement among annotators, calculated by the average of the maximum votes for each question, is 3.16 out of 5 (63.2%). The balance within the dataset indicates that while there is consensus on certain annotations, there is also a substantial variety in the answers within each country, reflecting a diverse range of perspectives. We also present the average number of annotations per question in Table 8 in the Appendix, to show the level of answer variance.

Table 9 in the Appendix presents the average number of *'I don't know'* responses per question. On average, there were 1.01 out of 5 such responses per question, with a standard deviation of 0.35 (ranging from a high of 1.912 in Northern Nigeria to a low of 0.42 in South Korea). The frequency of *'I don't know'* responses was higher in the *sports* and *holidays/celebrations/leisure* categories, likely due to questions on sports or holidays that are not widely recognized or celebrated in certain countries or regions.

Furthermore, we measure the overlap of answers between countries/regions by calculating the number of shared lemmas of the English versions of annotations to compare the trend between them and show the result in Figure 2. The result indicates that countries/regions with closely aligned cultural backgrounds exhibit higher overlaps in answers. The top pairs with the most similar responses are Indonesia & West Java (a province in Indonesia), the United States & the United Kingdom, and Spain & Mexico, likely due to shared historical, linguistic, or cultural ties that influence how questions are understood and answered. On the other hand, the pairs with the lowest value are Northern Nigeria & Greece/Ethiopia/South Korea. This could be due to the fact that Northern Nigeria has its own unique regional culture captured in the dataset.

## 4  LLMs Cultural Knowledge Evaluation

We measure how the current LLMs perform on BLEND on the two task settings: *short answer* and *multiple-choice*. Details for the experimental settings and the 16 evaluated models can be seen in Appendix C.1.

### 4.1  Short Answer Questions (SAQ)

**Experimental Setting.** In this experiment, we measure LLMs' performance on SAQ. The final score for each country is calculated as the average score over two prompts: 1) directly ask LLMs to provide the answer, and 2) add persona to the LLMs to make them act as a person from the target country or region. The detailed prompts are shown in Appendix C.2.1. To compute the score, we first mark the LLM's response as correct if it is included in the human annotators' responses to the same question. Then we compute the percentage of questions to which LLM's answer is correct. More details on calculating the scores can be found in Appendix C.2.2.

We compute the scores for all the countries based on the results obtained for the local language and English, respectively. We use lemmatizers and stemmers to handle highly inflectional languages such as Arabic and variations in words. The details are shown in Appendix C.2.2. In addition, we remove accents from words in languages that contain accents, such as Spanish and Greek, to ensure that the annotations from human annotators match the responses of LLMs. When computing the scores, we ignore questions for which three or more annotators do not know the answer.

#### 4.1.1  LLM Performance on SAQ

Figure 3a presents the performance of five LLMs on short answer questions in the local languages of target countries/regions. Table 10 shows the performance of all 16 LLMs evaluated. The results indicate a consistent trend of lower performance for lower resource languages [12].

Highlighting just a few results, the average LLM performance for US, Spain, Iran, North Korea, Northern Nigeria, and Ethiopia are 79.22%, 69.08%, 50.78%, 41.92%, 21.18%, and 12.18%, respectively, indicating a significant drop in performance for underrepresented cultures. Countries that share a common language but differ culturally show significant differences, for example, GPT-4, the highest-performing model, shows a substantial performance disparity of 31.63% between South Korea and North Korea. Similarly, between Spain and Mexico, GPT-4 exhibits a performance gap

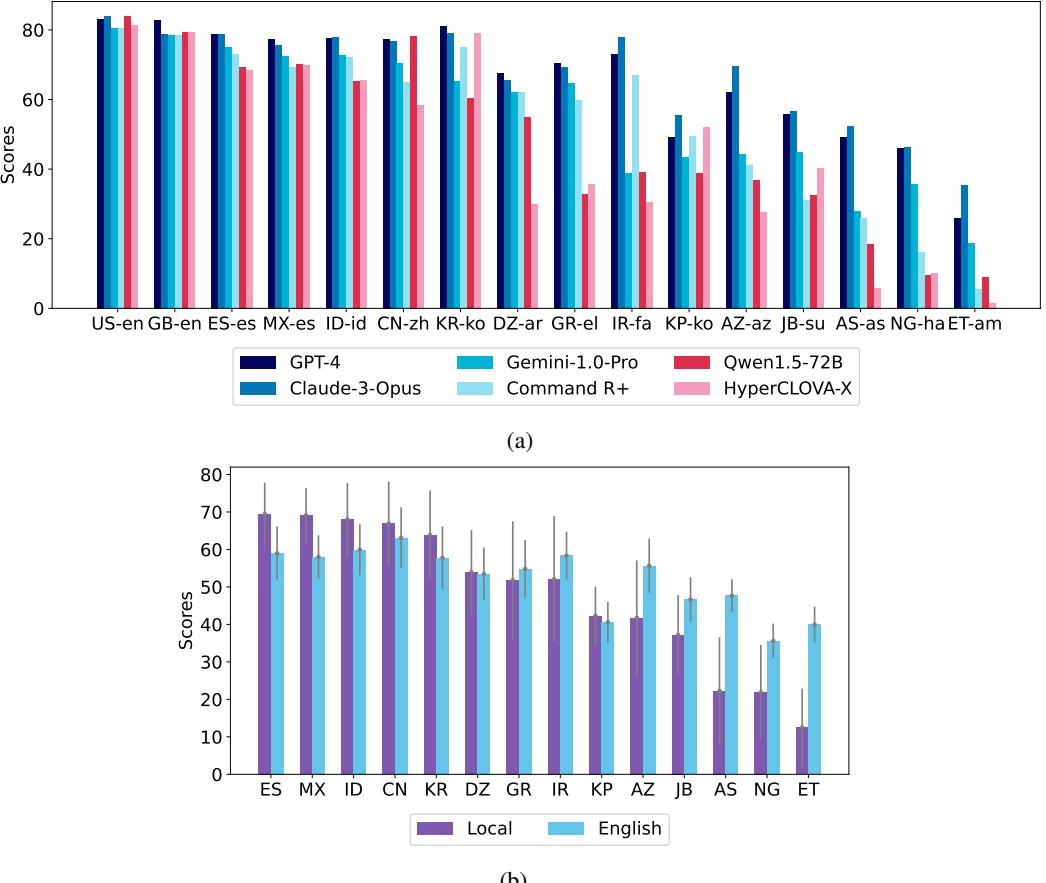

(a)

(b)

Figure 3: (a) LLMs' performance on short answer questions for each country/region in the local language. Models constructed from a Western country are shown in shades of blue, whereas those built from a non-Western country are shown in shades of red. (b) Average performance of all LLMs in local language and English on short answer questions. The grey error bars indicate the standard deviations among all models.

of 4.35%. Our findings highlight the critical need for LLMs to be trained on more diverse datasets, including low-resource languages and underrepresented cultures.

**Performance of Region-Centric LLMs.** Models built from non-Western countries tend to show higher performance on that specific country/region. For example, as seen in Figure 3a, Qwen1.5-72B [5], made by the Qwen Team in Alibaba [6] Group, shows highest performance on Chinese among all models. HyperCLOVA-X [34], built from the NAVER [7] HyperCLOVA Team, also shows comparable results on Korean, even exceeding GPT-4 performance in North Korean cultural questions. These language/region-specific models often benefit from customized datasets richer in local cultural content and nuances, typically underrepresented in the more universally used datasets, leading to higher performances in their regions.

**Local language vs. English.** We compare the average LLM performance when prompted in local languages versus English, as shown in Figure 3b [8]. For cultures represented by high-resource languages like Spanish and Chinese, the local languages show better performance across all models. In contrast, in cultures represented by low-resource languages such as Azerbaijani, Sundanese, and Amharic, English results in better performance (full results are shown in Table 11). This implies that the models' proficiency in a particular language significantly influences its performance and that models tend to show better cultural sensitivity in the local language when they possess sufficient

---

[6] Chinese technology company (`https://www.alibabagroup.com/`)

[7] Korean technology company (`https://www.navercorp.com/`)

[8] Performance on the six models presented in Figure 3a on the English version of SAQ is shown in Figure 13.

linguistic capability. Note for North Korean (KP) cultural questions, both English and Korean show poor performance as expected, but Korean performs slightly better, as it is a relatively high-resource language.

**Performance by Question Category.** In our analysis of six socio-cultural categories, models generally exhibit lower performance on questions related to *food* and *holidays/celebrations/leisure* than those concerning *work-life* or *education*. This disparity, significant with a $p < 0.05$ using one-way ANOVA, is detailed in Figure 15. This pattern indicates that more subjective topics like food and leisure are more challenging for LLMs to show cultural adaptiveness.

## 4.2 Multiple-Choice Questions (MCQ)

While SAQ is effective for multilingual evaluation, LLMs often generate responses that deviate from the annotators' one- or few-word answers, for example, generating long sentences, especially in languages that do not follow the instructions well. Hence we make the MCQ to enable simpler evaluation of LLMs. One limitation of our MCQ is that it is only available in English, as the incorrect options were chosen from different cultures' responses to the same questions, and translating all of those requires additional work. We plan to release a multilingual version of MCQ soon.

### 4.2.1 MCQ Construction

We make the multiple-choice questions about each target country/region in English, with other answer options from other countries/regions. For fair comparison across all countries, we remove questions for which at least one country has an annotation of *'not applicable to our culture,'* or more than three annotators don't know the answer. We also remove questions where all annotations have one vote each, indicating no typical answer from that country for that question. We determine the correct answer for each question by selecting the annotation with the highest votes from each country. We provide four answer options for each question, with no more than one option from any of the other countries. The detailed process of choosing plausible incorrect answer options can be seen in Appendix C.3.1. The final multiple-choice question prompt is shown in Appendix C.3.3.

### 4.2.2 LLM Performance on MCQ

In general, models show higher performance in MCQ than in SAQ as shown in Figure 4. This improvement is due to using questions with well-defined answers for multiple-choice questions. However, the pattern of displaying higher performance in high-resource cultures remains consistent. When considering the tendencies of all countries/regions for each model, the average Pearson correlation between the average performance in SAQ in the local languages and English across all countries/regions and the MCQ performance across all countries/regions is notably strong at 0.93. Furthermore, the Pearson correlation between the average model performance in English SAQ for all countries and that in MCQ exhibits a considerably high value of 0.98. This indicates a strong alignment between the two evaluation formats.

## 5 Human Evaluation

We conduct a human evaluation for short-answer responses from LLMs to understand the source of errors. We use responses from GPT-4, the best-performing model, for short-answer questions. We define the following categories: *stereotypical*, *partially correct*, *refusal*, *nonsensical*, *unnatural language*, and *different country's view* to analyze 120 wrong answers based on the automated evaluation. The detailed instructions and the definitions of each category can be found in Appendix D.3.1. Also, the summary of the human evaluation results can be found in Table 13.

The most stereotypical responses came from answers generated for underrepresented languages/cultures such as Ethiopia, West Java, and Assam, with 48.33% of responses from Ethiopia being stereotypical. Most stereotypical questions were related to food or festivals, where the LLM attempted to provide traditional information about the country or the region without fully understanding the context. For instance, for West Java, the LLM frequently answered any food-related questions with 'Seblak,' one of the most famous dishes originating from the region.

Notably, countries with a high percentage of partially correct answers or refusals were all from underrepresented cultures, such as Azerbaijan, North Korea, Northern Nigeria, and Ethiopia. This

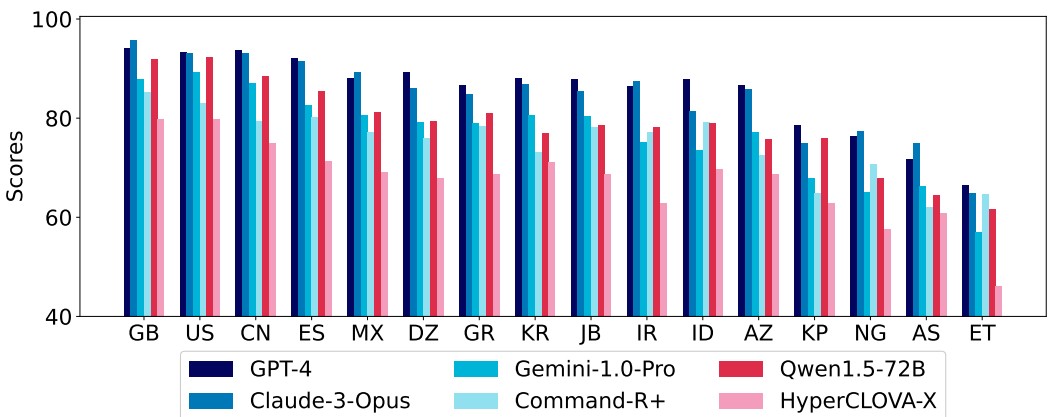

Figure 4: LLMs' performance on multiple-choice questions. Models constructed from a Western country are shown in shades of blue, whereas those built from a non-Western country are shown in shades of red. Similar to the results from short-answer questions, models tend to show lower performance in underrepresented countries/regions.

indicates that the LLMs tend to provide a long list of multiple answers or even refuse to answer when there is insufficient information about the topic/question. The same trend was observed for nonsensical answers, indicating that the capability of LLMs to comprehend questions is limited for low-resource languages. There were also many hallucinations for low-resource languages, such as providing 'Ruslan Cfrov' as the most famous basketball player in Azerbaijan, despite the non-existence of a famous player with that name.

GPT-4 also tends to provide answers from the perspective of other countries when responding to queries about Azerbaijan and North Korea. For Azerbaijan, many answers were from the perspectives of other countries in the Caucasus region, and for North Korea, most responses were from the perspective of South Korea. This aligns with the annotations for unnatural language, as the same two countries had the highest ratio of unnatural language. In the case of Azerbaijan, there were instances where the LLM even responded in Turkish. For North Korea, a surprising 18.33% of the responses were marked as unnatural because they were phrased in the words used exclusively in South Korea.

# 6 Conclusion

In this paper, we present BLEND, a benchmark to evaluate the cultural knowledge about everyday life within 16 current LLMs in 16 countries/regions and 13 distinct languages.

Our experimental findings indicate that current LLMs demonstrate a high level of competence in highly represented cultures such as the United States and the United Kingdom. However, their performance is significantly lower in the case of less-represented and underrepresented cultures and languages, especially when prompted in the local language. This outcome is observed in both short-answer questions and multiple-choice questions. Furthermore, our study reveals the performance gap between two countries using the same language, highlighting a cultural bias among those regions. Moreover, the study shows that the performance of LLMs varies depending on the language used in prompting: LLMs generally perform better in local languages for mid-to-highly represented cultures, while for underrepresented cultures, they perform better in English.

# 7 Limitations and Future Work

One limitation of our approach is the relatively small number of annotators, typically five per question, sometimes from the same locality within one country. This might not fully represent the countries/regions we include in our dataset. Extending efforts to increase the number of annotators per country, especially from diverse regional bases within each of the countries/regions, will be the most immediate future work of this research. Moreover, most language experts involved in the benchmark creation were academics proficient in English, the reference language for communication and translation. This may bias part of the construction process as they may not be fully representative

of the population of each country. We do not claim that our data fully represents all the speakers of any language/region, but our dataset remains a good starting point for researchers interested in the topic.

Additionally, evaluating short-answer questions poses noticeable challenges. Despite the extensive human effort and using lemmatizers/stemmers, accounting for all word variations is difficult, leading to correct answers not being evaluated accurately. Our dataset also faces challenges in evaluating long-form responses from LLMs, as the annotated data is based on short answers. Future work should focus on accurately evaluating the cultural adaptiveness of LLMs in long-form natural contexts, as limitations exist within prompt-based evaluations.

## Acknowledgments and Disclosure of Funding

This project was funded by the KAIST-NAVER hypercreative AI center. Alice Oh is funded by Institute of Information communications Technology Planning Evaluation (IITP) grant funded by the Korea government(MSIT) (No. 2022-000184, Development and Study of AI Technologies to Inexpensively Conform to Evolving Policy on Ethics). Moreover, this research project has benefitted from the Microsoft Accelerate Foundation Models Research (AFMR) grant program through which leading foundation models hosted by Microsoft Azure along with access to Azure credits were provided to conduct the research. Jose Camacho-Collados, Dimosthenis Antypas, and Yi Zhou are supported by a UKRI Future Leaders Fellowship.

We also thank the following annotators for their invaluable help in building the dataset: Ángela Collados Ais, Kiamehr Rezaee, Nurul Ariyani, Sabrina Borrelli, Trapsilo Bumi, Helia Taheri, Chao Tan, Guanqun Cao, Dimitra Mavridou, Abderrahmane Samir Lazouni, Noufel Bouslama, Lyes Taher Khalfi, Nitumoni Neog, Bhagyashree Deka, Sikha Swarnakar, Sangeeta Neog, Nitashree Neog, Hailegnaw Tilaye, Amare Lakew, Wasihun Lakew, Yohannes Bogale, Addis Alemayehu, Yeon Su Park, Hee Su Park, Jeong Min Young, Hyewon Im, Geunsoo Lee, David Chong, Dea Adhista, Sarah Oktavianti, Muhammad Syahrul Kurniawan, Taufik Muhamad Yusup, Miguel Ramirez, Thomas Welsch, annotators from Prolific, and all other annotators who preferred to remain unnamed.

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

# Appendix

## A Dataset Details

### A.1 Accessibility, Usage, License, and Maintenance

**Accessibility:** All data samples of BLEnD—including short answer questions, multiple-choice questions, and their answers—as well as the codes we use in our work, can be found at `https://github.com/nlee0212/BLEnD`. We also make our dataset publicly available at HuggingFace Datasets (`https://huggingface.co/datasets/nayeon212/BLEnD`).

**Usage:** In the GitHub repository, all the data samples for short-answer questions, including the human-annotated answers, can be found in the `data/` directory. Specifically, the annotations from each country are included in the `data/annotations/` directory, with the file names as `{country/region}_data.json`. Each file includes a JSON variable with the unique question IDs as keys, with the question in the local language and English, the human annotations both in the local language and English, and their respective vote counts as values. The example of an instance in the dataset for South Korea is shown below:

```
"A1-en-06": {
    "question": "대한민국 학교 급식에서 흔히 볼 수 있는 음식은 무엇인가요?",
    "en_question": "What is a common school cafeteria food in your country?",
    "annotations": [
        {
            "answers": [
                "김치"
            ],
            "en_answers": [
                "kimchi"
            ],
            "count": 4
        },
        {
            "answers": [
                "밥",
                "쌀밥",
                "쌀"
            ],
            "en_answers": [
                "rice"
            ],
            "count": 3
        },
        ...
    ],
    "idks": {
        "idk": 0,
        "no-answer": 0,
        "not-applicable": 0,
        "others": []
    }
},
```

We also include the prompts that we used for LLM evaluation in local languages and English in the data/prompts/ directory. Each file is named `{country/region}_prompts.csv`. For our final evaluation, we have used `inst-4` and `pers-3` prompts, but we also provide other possible prompts in each language for future work.

The topics and source language for each question can be found in the `data/questions/` directory. Each file is named `{country/region}_questions.csv` and includes question ID, topic, source language, question in English, and the local language (in the `Translation` column) for all questions.

The code for retrieving answers from LLMs for the short-answer questions is provided at `model_inference.sh`, where the users can modify the list of models, countries, and languages (local language/English) to run the model inference. The results of each model's inference on the questions will be saved in default at `model_inference_results/` directory. To calculate the scores for the short-answer questions, the users can run `evaluation/evaluate.sh`.

The multiple-choice questions and their answers can be found at `evaluation/mc_data/mc_questions_file.csv`. Multiple-choice questions and answers are generated through the codes found at `evaluation/multiple_choice_generation.sh`.

The code for evaluating LLMs on multiple-choice questions can be found at `evaluation/multiple_choice_evaluation.sh`, where the users can modify the list of models to evaluate. Users must input their API keys within these files for the required models for all evaluations.

**License:** CC BY-SA 4.0

**Maintenance:** On GitHub, we plan to continually update our code and constantly resolve any bugs and issues. We encourage contributions from community members and researchers.

## A.2 Country/Region & Language Codes

Table 3 shows the two-letter ISO codes for each country/region and local language. We use the codes throughout the main content of the paper and the supplementary materials.

Table 3: Two-letter ISO codes for each country/region and the corresponding local languages.

| Country/Region | Code | Language | Code |
|---|---|---|---|
| United States | US | English | en |
| United Kingdom | GB | | |
| China | CN | Chinese | zh |
| Spain | ES | Spanish | es |
| Mexico | MX | | |
| Indonesia | ID | Indonesian | id |
| South Korea | KR | Korean | ko |
| North Korea | KP | | |
| Greece | GR | Greek | el |
| Iran | IR | Persian | fa |
| Algeria | DZ | Arabic | ar |
| Azerbaijan | AZ | Azerbaijani | az |
| West Java | JB | Sundanese | su |
| Assam | AS | Assamese | as |
| Northern Nigeria | NG | Hausa | ha |
| Ethiopia | ET | Amharic | am |

## A.3 Annotation Examples

The examples of annotations for cultural questions within each topic (i.e., food, sport, family, education, holidays, and work-life) for each country/region in our dataset are shown in Figure 5, Figure 6, Figure 7, Figure 8, Figure 9, and Figure 10 respectively. All the answers are presented in both local languages and English.

## B Construction Details of BLEND

### B.1 Resource Availability of Languages

As illustrated in the main text, we select languages with varying levels of resource availability and recruit annotators who are native speakers of each language. The detailed resource availability of the languages included in BLEND is shown in Table 4.

| Question | Annotation | Country/Region |
|---|---|---|
| What street food do people from the US like to eat? | hot dogs: 4
hamburger: 1
tacos: 1
... | US |
| What street food do people from the UK like to eat? | kebabs: 2
burgers: 2
fish and chips: 2
... | UK |
| 中国人喜欢吃什么街头小吃？ | 烤肠 (roasted sausage): 3
烧烤 (barbecue): 2
糖葫芦 (candied haw): 1
... | CN |
| ¿Qué comida callejera les gusta comer a las personas de España? | churros (churros): 2
patatas fritas (French fries): 1
pipas (sunflower seeds): 1 | ES |
| ¿Qué comida callejera les gusta comer a las personas de México? | tacos (tacos): 5
quesadillas (quesadillas): 3
tamales (tamales): 2
... | MX |
| Makanan jalanan apa yang disukai oleh orang-orang dari Indonesia? | cilok (cilok): 3
bakso (meatball): 2
seblak (seblak): 1 | ID |
| 대한민국 사람들은 어떤 길거리 음식을 좋아하나요? | 떡볶이 (stir-fried rice cakes): 4
붕어빵 (bungeoppang): 1
델리만쥬 (delimanjoo): 1
... | KR |
| 북한 사람들은 어떤 거리 음식을 좋아 하나요? | 두부밥 (tofu rice): 4
인조고기밥 (synthetic meat rice): 2
김밥 (gimbap): 1
... | KP |
| Τι street food συνηθίζουν να τρώνε οι άνθρωποι στην Ελλάδα; | πιτόγυρο (pita gyro): 3
σουβλάκι (souvlaki): 1
πίτσα (pizza): 1 | GR |
| مردم در ایران چه غذاهای خیابانی دوست دارند بخورند؟ | فلافل (falafel): 2
سمبوسه (samosa): 1
پیراشکی (pastry): 1
... | IR |
| أي نوع من الأكلات الشعبية يحب الجزائريون تناولها؟ | الكسكس (couscous): 4
الشخشوخة (chakhchoukha): 2
الرشتة (rishta): 1
... | DZ |
| Azərbaycanlılar küçə yeməklərindən nə yeməyi xoşlayırlar? | dönər (doner kebab): 5 | AZ |
| Jajanan jalanan naon nu resep didahar ku urang Jawa Barat? | cilok (cilok): 2
baso (meatball): 2
mi hayam (chicken noodle):1
... | JB |
| অসমীয়া লোকে সাধাৰণতে কি ধৰণৰ ৰাস্তাৰ খাদ্য খোৱা পছন্দ কৰে? | ফুচকা (panipuri): 4
ম'ম (dumpling): 4
চাহ (tea): 1 | AS |
| Wane irin abincin titi ne mutanen Arewacin Najeriya suka fi son ci? | awara (fried bean cake): 3
gurasa(flatbread): 2
shinkafa (rice): 1
... | NG |
| ኢትዮጵያውያን ምን የጎዳና ምግብ ይወዳሉ? | ችፕስ (chips): 4
ቆሎ (qollo): 2 | ET |

Figure 5: Example annotations for a cultural question related to the topic of *food* for each country/region in our dataset. The questions and annotations are provided in different languages, with translations of the annotated answers into English included in brackets. Annotations are sorted in descending order based on the frequency (i.e., vote count) of an answer provided by annotators, each separated by a line break. The vote count for each answer is displayed as numbers.

| Question | Annotation | Country/Region |
|---|---|---|
| What is the most popular indoor sport in the US? | basketball: 5
hockey: 1 | US |
| What is the most popular indoor sport in the UK? | swimming: 2
netball: 2
badminton: 1
... | UK |
| 中国最受欢迎的室内运动是什么？ | 乒乓球 (table tennis): 3
羽毛球 (badminton): 2
电竞 (e-sports): 1 | CN |
| ¿Cuál es el deporte de interior más popular en España? | baloncesto (basketball): 2
futbol sala (indoor football): 2
fútbol 7 (7-a-side football): 1
... | ES |
| ¿Cuál es el deporte de interior más popular en México? | basquetbal (basketball): 3
natación (swimming): 1
box (boxing): 1
... | MX |
| Apa olahraga dalam ruangan yang paling populer di Indonesia? | bulutangkis (badminton): 4
futsal (futsal): 2
ping pong (table tennis): 1
... | ID |
| 대한민국에서 가장 인기 있는 실내 스포츠는 무엇인가요? | 클라이밍 (climbing): 2
배드민턴 (badminton): 1
농구 (basketball): 1 | KR |
| 북한에서 좋아 하는 실내 체육운동은 무엇인가요? | 탁구 (table tennis): 3
롱구 (basketball): 2
배구 (volleyball): 1
... | KP |
| Ποιο είναι το πιο δημοφιλές άθλημα εσωτερικού χώρου στην Ελλάδα; | μπάσκετ (basketball): 4
ποδόσφαιρο (football): 1 | GR |
| محبوب‌ترین ورزش سرپوشیده در ایران چیست؟ | والیبال (volleyball): 2
فوتسال (futsal): 2
بسکتبال (basketball): 1
... | IR |
| ما هي أشهر رياضة قاعة في الجزائر؟ | الملاكمة (boxing): 2
كرة اليد (handball): 1
كرة الطائرة (volleyball): 1
... | DZ |
| Azərbaycanda ən populyar qapalı idman növü hansıdır? | şahmat (chess): 3
basketbol (basketball): 1 | AZ |
| Naon olahraga jero rohangan nu pang populerna di Jawa Barat? | bulu tangkis (badminton): 4
futsal (futsal): 2
pingpong (table tennis):1
... | JB |
| অসমত কি সবাতোকৈ জনপ্ৰিয় ইনডোৰ ক্ৰীড়া কি? | লুডু (ludo): 4
কেৰম (carrom): 3
দবা (chess): 2
... | AS |
| Wanne wasan cikin gida da aka fi so a Arewacin Najeriya? | kwallon kafa (football): 1
kacici-kacici (riddle): 1 | NG |
| በኢትዮጵያ የትኛው ዓይነቱ የቤት ውስጥ ስፖርት በጣም ታዋቂ ነው? | idk (I don't know): 3
ቦክስ (boxing): 1 | ET |

Figure 6: Example annotations for a cultural question related to the topic of *sport* for each country/region in our dataset. The questions and annotations are provided in different languages, with translations of the annotated answers into English included in brackets. Annotations are sorted in descending order based on the frequency (i.e., vote count) of an answer provided by annotators, each separated by a line break. The vote count for each answer is displayed as numbers.

| Question | Annotation | Country/Region |
|---|---|---|
| What is a popular family activity with a child to do on weekends in the US? | go to a park: 2
bowling: 1
swim: 1
... | US |
| What is a popular family activity with a child to do on weekends in the UK? | go to the zoo: 2
go to the park: 2
walks: 1
... | UK |
| 在中国，周末和孩子一起做的一项受欢迎的家庭活动是什么？ | 去公园 (go to a park): 2
逛街 (shopping): 1
室外活动 (outdoor activities): 1
... | CN |
| ¿Cuál es una actividad familiar popular para hacer con un niño los fines de semana en España? | ir al parque (go to the park): 2
pasear (to walk): 2
jugar a videojuegos (play video games): 1
... | ES |
| ¿Cuál es una actividad familiar popular para hacer con un niño los fines de semana en México? | ir al parque (go to the park): 5
visitar a la abuelita (visit grandma): 1
ir al cine (go to the movies): 1 | MX |
| Apa kegiatan keluarga yang populer untuk dilakukan bersama anak pada akhir pekan di Indonesia? | jalan-jalan ke mall (going to the mall): 3
bersepeda (cycling): 2
nonton tv (watch tv): 1 | ID |
| 대한민국에서 주말에 아이와 함께하는 인기 있는 가족 활동은 무엇인가요? | 여행 (travel): 2
스포츠 (sports): 1
보드 게임 (board game): 1
... | KR |
| 북한에서 휴식일에 아이와 함께하는 많이 하는 가족 활동은 무엇인가요? | 사사끼 (card game): 1
장마당가기 (go to the market): 1
영화보기 (watching movie): 1
... | KP |
| Ποια είναι μια δημοφιλής οικογενειακή δραστηριότητα με ένα παιδί για τα σαββατοκύριακα στην Ελλάδα; | βόλτα (stroll): 1
κινηματογράφος (cinima): 1
παιδική χαρά (playground): 1 | GR |
| در ایران یک فعالیت خانوادگی محبوب با فرزند برای انجام دادن در آخر هفتهها چیست؟ | پیک نیک در پارک (picnic in the park): 1
سفر (travel): 1
مهمانی (party): 1
... | IR |
| ما هي النشاطات العائلية الشائعة التي يمكن القيام بها مع الأطفال في عطلة نهاية الأسبوع في الجزائر؟ | التنزه (hiking): 5 | DZ |
| Azərbaycanda həftə sonları ailə ilə birlikdə uşaqla nə etmək populyardır? | parklara getmək (go to parks): 3
oyun meydançalarına getmək (go to playgrounds): 1
bağ evinə getmək (go to the country house): 1
... | AZ |
| Naon kagiatan kulawarga anu populer dipigawe babarengan jeung budak pikeun dilakukeun dina ahir minggu di Jawa Barat? | olahraga (sports): 1
lalajo tipi (watching tv): 1
ngojay (swimming):1
... | JB |
| অসমত সপ্তাহান্তত শিশুসহ পৰিয়ালে কি জনপ্ৰিয় কাম কৰে? | ফুৰিব যায় (go for a walk): 3
গাৰ্দেনিং (gardening): 1
পিকনিকলৈ যায় (picnic): 1 | AS |
| Menene shahararren aikin gida da yara suka fi so suyi a karshen mako a Arewacin Najeriya? | shara (sweep): 3
wanki (washing): 1 | NG |
| በኢትዮጵያ በሳምንት መጨረሻ ቤተሰብ ከልጅ ጋር ለመስራት የታወቀ እንቅስቃሴ ምንድን ነው? | ሩጫ (running): 2
ልብስ ማጠብ (washing clothes): 1
ቤት ማጽዳት (house cleaning) | ET |

Figure 7: Example annotations for a cultural question related to the topic of *family* for each country/region in our dataset. The questions and annotations are provided in different languages, with translations of the annotated answers into English included in brackets. Annotations are sorted in descending order based on the frequency (i.e., vote count) of an answer provided by annotators, each separated by a line break. The vote count for each answer is displayed as numbers.

| Question | Annotation | Country/Region |
|---|---|---|
| What language is taught in schools in the US besides English? | spanish: 5
french: 3
german: 2
... | US |
| What language is taught in schools in the UK besides English? | french: 5
spanish: 3
german: 2 | UK |
| 在中国的学校里除了英语之外还教授哪种语言？ | 中文 (chinese): 4 | CN |
| ¿Qué idioma se enseña en las escuelas de España además del inglés? | francés (french): 5
latin (latin): 2
aleman (german): 1
... | ES |
| ¿Qué idioma se enseña en las escuelas de México además del inglés? | francés (french): 4
español (spanish): 2
nahuatl (nahuatl): 1
... | MX |
| Bahasa apa yang diajarkan di sekolah-sekolah di Indonesia selain Bahasa Inggris? | bahasa indonesia (indonesian): 2
mandarin (mandarin): 2
bahasa daerah (regional language): 1
... | ID |
| 대한민국의 학교에서 학생들은 영어 외에 어떤 언어를 배우나요? | 일본어 (japanese): 4
중국어 (chinese): 3
불어 (french): 1 | KR |
| 북한의 학교에서 학생들은 영어 외에 어떤 외국어를 배우나요? | 중국어 (chinese): 4
러시아어 (russian language): 3
한문 (chinese characters): 1 | KP |
| Ποια γλώσσα διδάσκεται στα σχολεία στην Ελλάδα πέρα από τα Αγγλικά; | γερμανικά (german): 5
γαλλικά (french): 5
ελληνικά (greek): 1 | GR |
| در ایران به جز انگلیسی، چه زبان‌هایی در مدارس تدریس داده می‌شود؟ | عربی (arabic): 4
انگلیسی (english): 1
فرانسه (france): 1
... | IR |
| أي لغة تُدرّس في المدارس الجزائرية بالإضافة إلى اللغة الإنجليزية؟ | الفرنسية (french): 5 | DZ |
| Azərbaycanda məktəblərdə ingilis dilindən başqa hansı dillər tədris edilir? | rus dili (russian): 5
alman dili (german): 2
fransız dili (french): 1 | AZ |
| Basa naon nu diajarkeun di sakola-sakola di Jawa Barat salian ti Basa Inggris? | basa indonesia (indonesian language): 4
basa sunda (sundanese language): 2
jepang (japanese language):2
... | JB |
| অসমৰ বিদ্যালয়সমূহত ইংৰাজীৰ উপৰিও আন কোন ভাষা শিক্ষা দিয়া হয়? | হিন্দী (hindi): 5
সংস্কৃত (sanskrit): 2
অসমীয়া (assamese): 2
... | AS |
| Wane yare ake koyarwa a makarantun Arewacin Najeriya banda Turanci? | hausa (hausa): 4
larabci (arabic): 4 | NG |
| በኢትዮጵያ ትምህርት ቤቶች ከእንግሊዝኛ ቋንቋ በተጨማሪ ምን ይማራል? | አማርኛ (amharic): 4
አሮምኛ (oromic): 1 | ET |

Figure 8: Example annotations for a cultural question related to the topic of *educate* for each country/region in our dataset. The questions and annotations are provided in different languages, with translations of the annotated answers into English included in brackets. Annotations are sorted in descending order based on the frequency (i.e., vote count) of an answer provided by annotators, each separated by a line break. The vote count for each answer is displayed as numbers.

| Question | Annotation | Country/Region |
|---|---|---|
| On which holiday do all family members tend to reunite in the US? | thanksgiving: 4
christmas: 2 | US |
| On which holiday do all family members tend to reunite in the UK? | christmas: 5 | UK |
| 在中国，哪个节日家里的所有成员会团聚？ | 春节 (spring festival): 4
中秋节 (mid-autumn festival): 4
清明 (qingming): 1 | CN |
| ¿En qué festivo suelen reunirse todos los miembros de la familia en España? | navidad (christmas): 3
nochebuena (christmas eve): 2
nochevieja (new year's eve): 2
... | ES |
| ¿En qué festividad suelen reunirse todos los miembros de la familia en México? | navidad (christmas): 5
año nuevo (new year): 3
16 de septiembre (september 16th): 1
... | MX |
| Pada hari libur apa semua anggota keluarga biasanya berkumpul di Indonesia? | idul fitri (eid al-fitr): 4
natal (christmas):3
tahun baru (new year): 2
... | ID |
| 대한민국에서 모든 가족 구성원들이 함께 모이는 명절은 무엇이 있나요? | 추석 (chuseok): 5
설날 (lunar new year): 5 | KR |
| 북한에서 모든 가족 식구들이 함께 모이는 명절은 무엇이 있나요? | 추석 (chuseok): 3
설날 (lunar new year): 2
양력설 (gregorian new year): 1
... | KP |
| Σε ποια εορτή συνηθίζουν όλα τα μέλη της οικογένειας να επανασυνδέονται στην Ελλάδα; | πάσχα (easter): 4
χριστούγεννα (christmas): 3
γενέθλια (birthday): 1 | GR |
| در ایران در کدام تعطیلات همه اعضای خانواده معمولاً دور هم جمع می‌شوند؟ | نوروز (new year): 4
چهارشنبه سوری (chaharshanbe suri): 1
سیزده بدر (nature's day): 1
... | IR |
| في أي عيد يجتمع أفراد العائلة في الجزائر؟ | عيد الفطر (eid al-fitr): 5
عيد الاضحى (eid al-adha): 4
رأس السنة (new year): 1 | DZ |
| Azərbaycanda ailə üzvləri hansı bayramda bir araya gəlirlər? | novruz bayramı (novruz): 5
yeni il bayramı (new year): 1 | AZ |
| Dina liburan naon sadaya anggota kulawarga biasana ngariung deui di Jawa Barat? | idul fitri (eid al-fitr): 4
libur lebaran (eid holiday): 1
natal (christmas):1
... | JB |
| অসমত কোন উৎসৱত সকলো পৰিয়ালৰ সদস্যসকল একত্ৰিত হ'বলৈ প্ৰৱণ হয়? | বিহু (bihu): 5
পূজা (puja): 1
দুৰ্গা পূজা (durga puja): 2 | AS |
| A wane hutun ne dukkan 'yan uwa sukan hadu a Arewacin Najeriya? | hutun sallah (eid holiday): 4
hutun kistimeti (christmas): 3 | NG |
| በኢትዮጵያ በየትኛው በዓል ሁሉም ቤተሰቦች በአንድ ላይ ለመሆን ይሻሉ? | ፋሲካ (easter): 2
ረመዳን (ramadan): 1
ዘመን መለወጫ (new year) | ET |

Figure 9: Example annotations for a cultural question related to the topic of *holiday* for each country/region in our dataset. The questions and annotations are provided in different languages, with translations of the annotated answers into English included in brackets. Annotations are sorted in descending order based on the frequency (i.e., vote count) of an answer provided by annotators, each separated by a line break. The vote count for each answer is displayed as numbers.

| Question | Annotation | Country/Region |
|---|---|---|
| What is regarded as the most important perk typically offered to employees in the US? | vacation: 3
healthcare: 3
benefits: 1
... | US |
| What is regarded as the most important perk typically offered to employees in the UK? | bonus: 2
free lunches: 1
pension: 1
... | UK |
| 在中国，通常认为给员工提供的最重要的福利是什么？ | 五险一金 (five insurances and one fund): 3
双休 (weekends off): 2
年假: annual leave: 1
... | CN |
| ¿Cuál se considera el beneficio más importante que se ofrece típicamente a los empleados en España? | la seguridad social (social security): 2
salario (salary): 1
tiempo libre (free time): 1
... | ES |
| ¿Cuál se considera el beneficio más importante que se ofrece típicamente a los empleados en México? | imss (mexican social security institute): 2
vacaciones pagadas (paid vacations): 2
afore (retirement fund administration companies): 1
... | MX |
| Apa yang dianggap sebagai keuntungan paling penting yang biasanya ditawarkan kepada karyawan di Indonesia? | gaji (salary): 3
thr (religious holiday allowance): 1
bonus tahunan (annual bonus): 1
... | ID |
| 대한민국에서 일반적으로 직원들에게 제공되는 혜택 중 가장 중요하게 여겨지는 것은 무엇인가요? | 보너스 (bonus): 2
직원가 할인 (employee discount): 2
휴가 (vacation): 1
... | KR |
| 북한에서 일반적으로 로동자들에게 주는 사회급양, 표창 및 휴양소 휴가 중 가장 중요하게 여기는 것은 무엇인가요? | 사회급양 (social distribution): 2
휴양소 휴가 (resort vacation): 1
표창 휴가 (commendation): 1 | KP |
| Ποιο θεωρείται το σημαντικότερο προνόμιο που συνήθως προσφέρεται στους εργαζομένους στην Ελλάδα; | ασφάλιση (insurance): 2
κοντινές διακοπές (short breaks): 1
άδεια (days off): 1 | GR |
| در ایران مهم ترین مزیتی که معمولاً به کارمندان ارائه می‌شود، چیست؟ | بیمه (insurance): 2
حقوق بازنشستگی (pension): 1
پاداش اضافه کار (overtime bonus): 1 | IR |
| ما هي أهم ميزة تُقدم عادةً للموظفين في الجزائر؟ | الراتب (salary): 2
علاوة (allowance): 2
سيارة وظيفة (official car): 1 | DZ |
| Azərbaycanda işçilərə adətən təklif edilən ən önəmli imtiyaz nə hesab olunur? | uzun məzuniyyət (long vacation): 1
rütbə artımı (promotion): 1
maaş (salary): 1 | AZ |
| Naon nu dianggap minangka kauntungan pang pentingna nu biasana ditawarkeun ka karyawan di Jawa Barat? | asuransi kasihata (health insurance): 2
gajih (salary): 1
bonus (bonus): 1
... | JB |
| অসমত কৰ্মচাৰীসকলক সাধাৰণতে দিয়া সবাতোকৈ গুৰুত্বপূৰ্ণ সুবিধাটো কি হিচাপে গণ্য কৰা হয়? | স্বাস্থ্য বীমা সুবিধা (health insurance benefit): 2
বিনামূলীয়া চিকিৎসা (free treatment): 1 | AS |
| Menene ake dauka a matsayin mafi muhimmancin alawus da ake bayarwa ga ma'aikata a Arewacin Najeriya? | kuɗi (money): 2 | NG |
| በኢትዮጵያ ለሠራተኞች ተለይቶ የሚቀርብ እና እጅግ ዋና የሆነ ተጨማሪ አበል ምንድነው? | የቤት አበል (housing allowance): 2
ውሎ አበል (allowance): 1
ቦነስ (bonus): 1 | ET |

Figure 10: Example annotations for a cultural question related to the topic of *work life* for each country/region in our dataset. The questions and annotations are provided in different languages, with translations of the annotated answers into English included in brackets. Annotations are sorted in descending order based on the frequency (i.e., vote count) of an answer provided by annotators, each separated by a line break. The vote count for each answer is displayed as numbers.

Table 4: Resource availability of the 13 languages covered in BLEND. The resource availability is defined by [12].

| Class | Languages |
|---|---|
| 1 - The Left-Behinds | Assamese, Azerbaijani, Sundanese |
| 2 - The Hopefuls | Amharic, Hausa |
| 3 - The Rising Stars | Greek, Indonesian |
| 4 - The Underdogs | Korean, Persian |
| 5 - The Winners | Arabic, Chinese (Mandarin), English, Spanish |

Table 5: Annotator demographics for each country or region who are recruited via Prolific.

| | US | GB | CN | ES | ID | GR | MX | IR |
|---|---|---|---|---|---|---|---|---|
| **No. of Annotators** | 87 | 119 | 59 | 91 | 40 | 86 | 86 | 50 |
| **Gender (%)** | | | | | | | | |
| Female | 42.53 | 46.22 | 55.93 | 49.45 | 50.00 | 45.35 | 48.84 | 56.00 |
| Male | 52.87 | 49.58 | 44.07 | 49.45 | 50.00 | 54.65 | 48.84 | 42.00 |
| Non-binary | 4.60 | 2.52 | - | 1.10 | - | - | 2.33 | 2.00 |
| Prefer not to say | - | 1.68 | - | - | - | - | - | - |
| **Age (%)** | | | | | | | | |
| -29 | 36.78 | 13.45 | 64.41 | 41.76 | 45.00 | 50.00 | 59.30 | 48.00 |
| 30-39 | 19.54 | 26.89 | 25.42 | 23.08 | 35.00 | 29.07 | 26.74 | 44.00 |
| 40-49 | 17.24 | 21.01 | 3.39 | 18.68 | 12.50 | 13.95 | 8.14 | 8.00 |
| 50-59 | 14.94 | 21.85 | 6.78 | 14.29 | 7.50 | 6.98 | 4.65 | - |
| 60+ | 11.49 | 16.81 | - | 2.20 | - | - | 1.16 | - |
| **Duration of Residence in Target Country (%)** | | | | | | | | |
| 100% | 55.17 | 75.63 | 1.69 | 75.82 | 5.00 | 86.05 | 75.58 | 8.00 |
| ≥ 90% | 9.20 | 7.56 | 28.81 | 10.99 | 25.00 | 1.16 | 16.28 | 34.00 |
| ≥ 80% | 13.79 | 5.04 | 23.73 | 5.49 | 20.00 | 6.98 | 2.33 | 22.00 |
| ≥ 70% | 6.90 | 3.36 | 15.25 | 5.49 | 17.50 | 5.81 | 4.65 | 20.00 |
| ≥ 60% | 9.20 | 5.04 | 25.42 | 2.20 | 12.50 | - | 1.16 | 10.00 |
| ≥ 50% | 5.75 | 2.52 | 5.08 | - | 20.00 | - | - | 6.00 |
| **Education Level (%)** | | | | | | | | |
| Below High School | - | 0.84 | - | 3.30 | - | - | - | 2.00 |
| High School | 11.49 | 12.61 | 6.78 | 12.09 | 20.00 | 13.95 | 15.12 | 4.00 |
| College | 22.99 | 21.85 | 3.39 | 16.48 | 2.50 | 11.63 | 4.65 | 10.00 |
| Bachelor | 47.13 | 48.74 | 35.59 | 40.66 | 30.00 | 40.70 | 66.28 | 32.00 |
| Master's Degree | 18.39 | 13.45 | 38.98 | 21.98 | 40.00 | 25.58 | 11.63 | 46.00 |
| Doctorate | - | 2.52 | 15.25 | 5.49 | 7.50 | 8.14 | 2.33 | 6.00 |

## B.2 Ethical Considerations of Annotator Recruitment

This research project was performed under approval from KAIST IRB (KH2023-226). We obtained 'Informed Consent for Human Subjects' from the annotators. We embedded the consent document within the annotation website for the crowdworkers or received written consent from the directly recruited annotators. The annotations were gathered only from those who had read and consented to the form. We recruited annotators without any discrimination based on age, ethnicity, disability, or gender. Workers were compensated at a rate exceeding Prolific's ethical standards [9]. These same standards were applied to workers directly recruited for the annotation of low-resource languages.

Participants could voluntarily decide to join or withdraw from the study, and any data provided would not be used for research purposes if they withdraw. Additionally, the annotators were notified that if an unexpected situation arises during participation, appropriate actions will be taken according to the situation, and documents complying with the requirements of the KAIST IRB will be promptly prepared and reported.

---

[9] https://www.prolific.com/resources/how-much-should-you-pay-research-participants

Table 6: Annotator demographics for each country or region who are recruited directly.

| | KR | DZ | AZ | KP | JB | AS | NG | ET |
|---|---|---|---|---|---|---|---|---|
| **No. of Annotators** | | | | 5 | | | | |
| **Gender (%)** | | | | | | | | |
| Female | 60.00 | 40.00 | 40.00 | 80.00 | 40.00 | 100.00 | 60.00 | - |
| Male | 40.00 | 60.00 | 60.00 | 20.00 | 60.00 | - | 40.00 | 100.00 |
| Non-binary | - | - | - | - | - | - | - | - |
| Prefer not to say | - | - | - | - | - | - | - | - |
| **Age (%)** | | | | | | | | |
| -29 | 60.00 | 20.00 | 100.00 | - | 100.00 | 60.00 | 60.00 | 60.00 |
| 30-39 | - | 60.00 | - | - | - | 40.00 | 40.00 | 40.00 |
| 40-49 | - | - | - | 40.00 | - | - | - | - |
| 50-59 | 40.00 | 20.00 | - | 60.00 | - | - | - | - |
| 60+ | - | - | - | - | - | - | - | - |
| **Duration of Residence in Target Country (%)** | | | | | | | | |
| 100% | 20.00 | 80.00 | - | - | 80.00 | 80.00 | 80.00 | 100.00 |
| $\geq 90\%$ | - | - | - | - | - | - | - | - |
| $\geq 80\%$ | 40.00 | - | 80.00 | 20.00 | - | - | 20.00 | - |
| $\geq 70\%$ | 20.00 | 20.00 | 20.00 | - | 20.00 | - | - | - |
| $\geq 60\%$ | 20.00 | - | - | - | - | - | - | - |
| $\geq 50\%$ | - | - | - | 20.00 | - | 20.00 | - | - |
| $< 50\%$ | - | - | - | 60.00 | - | - | - | - |
| **Education Level (%)** | | | | | | | | |
| Below High School | - | - | - | - | - | - | - | - |
| High School | 60.00 | - | 80.00 | - | 40.00 | - | 20.00 | - |
| College | - | - | - | 20.00 | - | - | - | - |
| Bachelor | 40.00 | 40.00 | 20.00 | 20.00 | 60.00 | 20.00 | 60.00 | 20.00 |
| Master's Degree | - | 40.00 | - | 60.00 | - | 80.00 | 20.00 | 80.00 |
| Doctorate | - | 20.00 | - | - | - | - | - | - |

## B.3  Annotator Demographics

The statistics of all annotators participating in our dataset construction are shown in Table 5 and 6.

Table 7: Average of maximum votes among all answers for each question in different categories across countries. A value of '3.00' indicates that, on average, three annotators provided the same answer for each question.

| Category | US | GB | ES | MX | ID | CN | KR | DZ | GR | IR | KP | AZ | JB | AS | NG | ET |
|---|---|---|---|---|---|---|---|---|---|---|---|---|---|---|---|---|
| Food | 3.12 | 3.14 | 2.99 | 3.27 | 2.93 | 2.67 | 3.28 | 3.29 | 2.91 | 2.99 | 2.61 | 3.19 | 3.01 | 3.14 | 2.72 | 3.04 |
| Sport | 3.35 | 3.47 | 3.57 | 3.53 | 3.59 | 3.07 | 3.57 | 3.09 | 3.30 | 3.59 | 2.89 | 3.24 | 3.47 | 2.97 | 2.98 | 3.18 |
| Family | 3.17 | 3.40 | 3.17 | 3.16 | 3.16 | 3.08 | 3.40 | 2.94 | 3.19 | 3.17 | 2.81 | 3.25 | 2.94 | 3.19 | 2.65 | 2.78 |
| Education | 3.24 | 3.26 | 3.30 | 3.19 | 3.21 | 3.25 | 3.63 | 3.18 | 3.29 | 3.20 | 3.27 | 3.42 | 3.45 | 3.10 | 2.94 | 3.23 |
| Holidays | 3.09 | 3.33 | 3.18 | 3.28 | 3.14 | 3.04 | 3.60 | 3.04 | 2.98 | 3.20 | 3.07 | 3.27 | 3.10 | 2.92 | 2.60 | 3.12 |
| Work-life | 3.10 | 3.19 | 3.09 | 3.00 | 3.22 | 3.15 | 3.57 | 3.31 | 2.87 | 3.09 | 3.01 | 3.59 | 3.10 | 3.25 | 2.75 | 3.12 |
| **Overall** | **3.18** | **3.29** | **3.22** | **3.25** | **3.20** | **3.02** | **3.50** | **3.15** | **3.08** | **3.21** | **2.93** | **3.31** | **3.18** | **3.08** | **2.78** | **3.09** |

## B.4 Question Construction Guidelines

Below are the annotation guidelines for creating the question templates in BLEND.

> The goal of this task is to write question-and-answer pairs that ask about your country's culture. In each spreadsheet, you need to write down the questions and the corresponding answers to each question. Write them down in your native language, and add their translation into English too in the spreadsheet provided.
>
> Please find below a few guidelines to take into account when writing the questions:
>
> - **Questions and answers should be a culture specific question related to your culture** (can be a common sense question). For example, a question related to the sport topic could be "What is the most popular sport in your country?". You should refrain from writing factual questions as much as possible.
>
> - **Do not generate yes or no questions or answers that only have two options** (e.g. male or female). You could convert a yes or no question to a question starting with question words. Instead of asking "'Do people in your country tend to get off work at 5:30 pm?", you may ask "What time do people in your country tend to get off work?".
>
> - **Please write questions distinct from each other as much as possible** under each topic.
>
> - **The answer should be short and concrete**. It is better to use precise concepts, entities, time, etc. to answer each question.
>
> - **Please avoid asking questions about a very stereotypical topic**. For instance, avoid questions like "Who bears more responsibility for taking care of children at home in your country?"

## B.5 Answer Annotation Guidelines

Figure 11 shows the annotation guidelines given to the annotators for all countries/regions. We provided guidelines, all in their local languages.

## B.6 Answer Annotation Interface

Figure 12 shows the annotation interface shown to the crowdworkers annotators in Prolific. We used an Excel sheet for annotators recruited by direct recruitment for the annotations (i.e., for low-resource languages).

## B.7 Annotation Analysis

Table 7 shows the level of agreement between the annotators, calculated by averaging the maximum votes among answers for each question in different categories across countries. Additionally, Table 8 shows the average number of answers per questions per categories across countries. Lastly, Table 9 shows the average number of *I don't know* per questions per categories across countries.

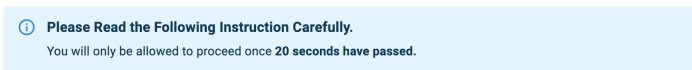

Figure 11: Answer annotation guidelines shown to the annotators.

Table 8: Average number of annotations for each question in different categories across countries. A value of '3.00' indicates that, on average, three answers were provided as the answer for each question.

| Category | US | GB | ES | MX | ID | CN | KR | DZ | GR | IR | KP | AZ | JB | AS | NG | ET |
|---|---|---|---|---|---|---|---|---|---|---|---|---|---|---|---|---|
| Food | 4.93 | 4.40 | 4.80 | 5.36 | 5.03 | 4.64 | 3.48 | 3.15 | 4.54 | 4.53 | 4.21 | 3.30 | 3.94 | 5.20 | 3.23 | 3.02 |
| Sport | 4.06 | 3.82 | 3.60 | 3.49 | 3.72 | 4.13 | 2.72 | 2.13 | 3.25 | 3.16 | 3.58 | 2.14 | 2.55 | 3.90 | 2.16 | 2.00 |
| Family | 4.41 | 3.44 | 3.71 | 4.78 | 4.32 | 3.81 | 2.84 | 2.38 | 3.38 | 3.43 | 3.60 | 2.86 | 2.48 | 4.46 | 2.63 | 2.59 |
| Education | 3.93 | 3.23 | 3.49 | 3.90 | 3.89 | 3.57 | 2.81 | 2.55 | 3.32 | 3.25 | 3.52 | 2.71 | 3.11 | 4.74 | 2.94 | 2.49 |
| Holidays | 4.40 | 3.62 | 3.77 | 4.40 | 4.15 | 4.04 | 2.41 | 2.42 | 3.57 | 3.41 | 3.20 | 2.46 | 3.12 | 5.14 | 2.49 | 2.57 |
| Work Life | 4.44 | 3.93 | 3.71 | 4.44 | 4.28 | 4.10 | 2.54 | 2.84 | 3.63 | 3.84 | 3.60 | 2.49 | 3.09 | 4.21 | 2.74 | 2.56 |
| **Overall** | **4.38** | **3.77** | **3.89** | **4.41** | **4.25** | **4.08** | **2.83** | **2.60** | **3.66** | **3.64** | **3.64** | **2.67** | **3.10** | **4.65** | **2.71** | **2.55** |

# C Experimental Settings for LLM Evaluation

## C.1 Models

We use GPT-4 (gpt-4-1106-preview) [24], GPT-3.5 (gpt-3.5-turbo-1106) [10], Claude-3-Opus (claude-3-opus-20240229), Claude-3-Sonnet (claude-3-sonnet-20240229), Claude-3-Haiku (claude-3-haiku-20240307) [11], Llama-3.1-70B (Llama-3.1-70B-Instruct) [12], PaLM2 (text-bison-002) [2], Gemini-1.0-Pro [30], C4AI Command R+ [13], C4AI Command R [14], Qwen-

---

[10] https://platform.openai.com/docs/models

[11] https://www.anthropic.com/api

[12] https://huggingface.co/meta-llama/Llama-3.1-70B-Instruct

[13] https://docs.cohere.com/reference/about

[14] https://huggingface.co/CohereForAI/c4ai-command-r-v01

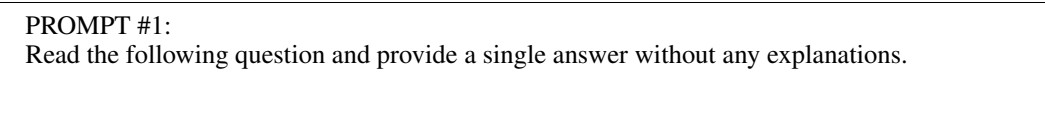

## Cultural Questions (1/30)

Answer the following question based on your **cultural perspective**. Make sure that your answer is in your **native language,** and is **short/concrete.**

**Question:**

Who is the most famous track and field athlete in the US?

**Answer:**

| Answer 1 |
| Answer 2 (Optional) |
| Answer 3 (Optional) |

**If you don't know the answer for the question, please select the reason from below.**

☐ There is no specific answer for this question.

☐ This question doesn't apply to our culture.

☐ This question may apply to our culture, but I don't know the answer.

☐ Others

‹ PREV                                                                 NEXT ›

Figure 12: Annotation interface given to the annotators.

Table 9: Average number of *I don't know* for each question in different categories across countries. A value of '1.00' indicates that, on average, one of the annotators failed to provide the answer to the question.

| Category | US | GB | ES | MX | ID | CN | KR | DZ | GR | IR | KP | AZ | JB | AS | NG | ET |
|---|---|---|---|---|---|---|---|---|---|---|---|---|---|---|---|---|
| Food | 0.80 | 0.71 | 0.49 | 0.36 | 0.56 | 0.68 | 0.23 | 1.00 | 0.52 | 0.90 | 1.24 | 0.58 | 1.15 | 0.54 | 1.69 | 0.81 |
| Sport | 1.58 | 1.70 | 1.22 | 1.11 | 1.08 | 0.92 | 0.31 | 1.65 | 1.41 | 1.67 | 1.44 | 1.64 | 1.69 | 1.01 | 2.16 | 1.53 |
| Family | 1.24 | 1.24 | 1.03 | 0.51 | 0.92 | 0.81 | 0.40 | 1.33 | 1.13 | 1.29 | 1.30 | 0.71 | 1.59 | 0.63 | 2.13 | 1.16 |
| Education | 0.92 | 1.02 | 0.83 | 0.39 | 0.48 | 0.37 | 0.24 | 0.82 | 0.58 | 0.52 | 0.51 | 0.37 | 0.69 | 0.25 | 1.42 | 0.61 |
| Holidays | 1.42 | 1.50 | 1.33 | 0.71 | 0.68 | 1.23 | 0.88 | 1.91 | 1.24 | 1.38 | 1.80 | 1.25 | 1.47 | 0.93 | 2.48 | 1.10 |
| Work Life | 0.71 | 1.10 | 0.91 | 0.63 | 0.43 | 0.69 | 0.49 | 0.62 | 1.13 | 1.16 | 1.29 | 0.60 | 1.22 | 0.63 | 1.59 | 0.68 |
| **Total** | **1.11** | **1.20** | **0.95** | **0.62** | **0.69** | **0.79** | **0.42** | **1.24** | **0.98** | **1.15** | **1.27** | **0.87** | **1.29** | **0.67** | **1.91** | **0.98** |

1.5-72B/32B/14B-Chat [5], SeaLLM-7B-v2.5 [23], HyperCLOVA X [34], Aya-23 [4] and Aya-101 [37] for evaluation of cultural sensitivity of LLMs.

8 Quadro RTX 8000 48GB were used with CUDA version 11.8 for all experiments. For all models, we use greedy decoding (temperature=0, top_p=1.0). We use the PyTorch library [15] for all experiments.

## C.2 Short Answer Question

### C.2.1 Prompt for Short Answer Evaluation

We use two types of prompts to evaluate the LLMs in short answer settings: 1) directly ask LLMs to provide the answer and 2) add persona to the LLMs to make them act as a person from the target country or region. The details of each prompt are as follows:

PROMPT #1:
Read the following question and provide a single answer without any explanations.

---

[15]https://pytorch.org/

Question: {QUESTION}
Answer:

PROMPT #2:
You are a person from {COUNTRY/REGION} who is trying to explain your country's culture to a foreigner. Answer the following question, providing a single answer without any explanations.

{QUESTION}

### C.2.2 Details of Short Answer Evaluation

Let $Q$ denote the question set, $A_q$ the annotated answer set for each question $q \in Q$, with each answer $a \in A_q$, for a question $q$ in the country or region $c$ in the human annotation. For any LLM prediction $y$, we define $s_{q,c}(y)$ as

$$s_{q,c}(y) = \begin{cases} 1, & \text{if } \exists a \in A_q \text{ such that } a \subseteq y \\ 0, & \text{otherwise} \end{cases} \tag{1}$$

so that $s_{q,c}(y)$ is 1 if the prediction $y$ includes any of the answers from the human annotations, denoted as $a \subseteq y$, and 0 otherwise. For a model $m$ that outputs $f_m(q, c)$ when given $q$ and $c$, the score $S(c)$ for each country or region $c$ is calculated as

$$S(c) = \frac{1}{|Q|} \sum_{q \in Q} s_{q,c}(f_m(q, c)) \times 100. \tag{2}$$

To evaluate LLM responses, we lemmatize/stem/tokenize the annotations and LLM responses for each question to consider the language variations. We use one of the three techniques that are available for each language.

We use the lemmatizer from the English model from SpaCy (`en_core_web_sm`) for English. For Spanish and Amharic, we use lemmatizers from SparkNLP [16]. For Indonesian, we use the lemmatizer from Kumparan NLP Library [17]. For Chinese, we use jieba [18], a Chinese word segmentation module. For Korean, we use the Okt lemmatizer from the konlpy package [19]. For Arabic, we use Qalsadi Arabic Lemmatizer [35]. For Greek, we use the CLTK Greek lemmatizer [11]. For Persian, we use Hazm, a Persian NLP Toolkit [20]. For Azerbaijani, we use the Azerbaijani Language Stemmer [21]. We use SUSTEM, a Sundanese Stemmer [27] for Sundanese. We use the Assamese tokenizer from Indic NLP Library [16] for Assamese. For Hausa, we use the Hausa Stemmer [6].

## C.3 Multiple Choice Question

### C.3.1 Multiple Choice Question Construction

To create plausible incorrect answer options for questions about the target country/region, we first consider all answer annotations from all other countries with at least two votes. Then, we sort these answer candidates by their vote count from each country/region. Next, we check each candidate to see if it is similar to any annotations collected from the target country/region. If it is, we block that candidate from being added as a wrong answer choice, as well as the same answer from the other countries/regions. We use GPT-4 to determine if two words are similar in meaning, such as 'fruit' and 'apple', as the two can be considered the same when answering the question. The prompt can be seen in Appendix C.3.2.

As this process would lead to differing possible wrong answer options for each target country per question, we pick the answer options with the minimum number of possible wrong answer options among all countries. If there are $n$ possible answer choices, we include all combinations of $\binom{n}{3}$ if

---

[16]Spanish lemmatizer (https://sparknlp.org/2020/02/16/lemma_es.html), Amharic lemmatizer (https://sparknlp.org/2021/01/20/lemma_am.html)

[17]https://github.com/kumparan/nlp-id/tree/v0.1.9.9

[18]https://github.com/fxsjy/jieba?tab=readme-ov-file

[19]https://konlpy.org/en/latest/api/konlpy.tag/

[20]https://github.com/roshan-research/hazm

[21]https://github.com/aznlp-disc/stemmer

$n \geq 3$, or include all $n$ answer choices plus $3 - n$ dummy options otherwise. We use GPT-4 (see Appendix C.3.2 for the prompt details) to produce dummy answer options to make the number of options comprised of one correct answer and three wrong answer options four. If there are multiple correct answers, we generate multiple versions of the question, each with a different correct answer. The choices are provided in alphabetical order when asked to LLMs in a multiple-choice format.

### C.3.2 Prompt for Multiple Choice Question Construction

**Similar Term Detection.** Since we asked the human annotators to provide answers in a short answer format, there may be cases where different textual answers refer to the same meaning. To avoid duplicate options in multiple-choice format, we utilized GPT-4 to determine whether the answers have the same meaning using the following prompt:

> Determine if a 'target' word is the same in meaning(e.g., football & soccer or soccer & football) to at least one of the 'answer' words, or one is a subset to another(e.g., fruit & apple or apple & fruit). If so, the 'result' for 'target' word is 'O'. However, if the two simply falls into the same level of hierarchy, the 'result' is 'X' (banana & apple, rose & carnation).
>
> Note that the 'answer' list is from 'answer_country,' and the 'target' word is from 'target_country,' as written by a person.
>
> Write down your reasoning first. Do not write any other JSON formatted object in your answer except for the result JSON object, formatted as {"result":"O"} or {"result":"X"}.

**Dummy Options Generation.** In cases where a question has fewer than four options during the option generation process, we ask GPT-4 to produce dummy options using the following prompt:

> Provide $\{3 - n\}$ dummy option(s) that makes sense to be the answer(s) of the given "question", and has to exist in real-life (non-fiction), but is totally different from the given "answers" without any explanation. Make sure that the options are different from each other, and cannot be an answer from any country. Provide as JSON format: {"dummy_options":[]}

### C.3.3 Prompt for Multiple Choice Evaluation

We use the following prompt to evaluate the LLMs' performance in multiple-choice format:

> {QUESTION} Without any explanation, choose only one from the given alphabet choices(e.g., A, B, C). Provide as JSON format: {"answer_choice":""}
>
> A. {CHOICE 1}
> B. {CHOICE 2}
> C. {CHOICE 3}
> D. {CHOICE 4}
>
> Answer:

## D   Detailed LLM Performance Analysis

### D.1   LLM Evaluation Results

Figure 13 shows the performance of models presented in 3a in SAQ when asked in English. Table 10 and Table 11 show the performance of all LLMs experimented on SAQ for all countries/regions on the local language and English, respectively.

Table 12 shows the performance of all LLMs on MCQ for all countries/regions.

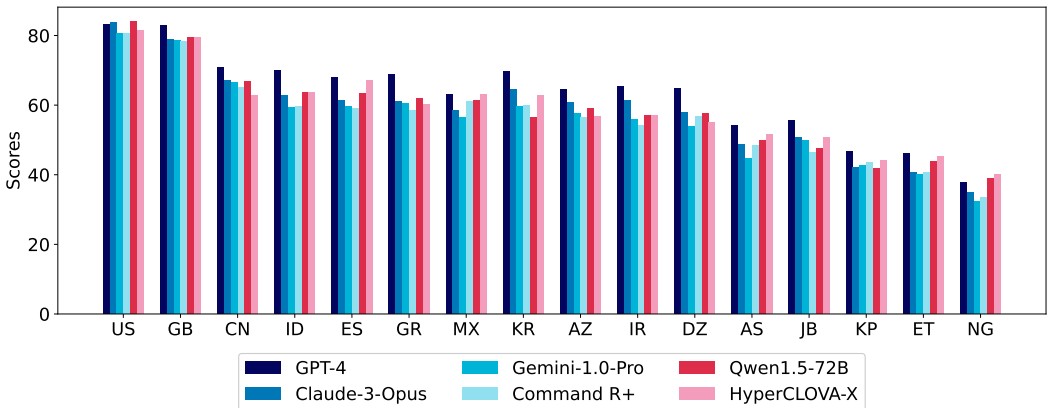

Figure 13: LLMs' performance on short answer questions for each country/region in English. Models constructed from a Western country are shown in shades of blue, whereas those built from a non-Western country are shown in shades of red.

Table 10: Performance of all LLMs on short answer questions for each country/region in local language.

| | US
en | GB
en | ES
es | MX
es | ID
id | CN
zh | KR
ko | DZ
ar |
|---|---|---|---|---|---|---|---|---|
| **GPT-4** | 83.19 | 82.75 | 79.00 | 77.45 | 77.50 | 77.32 | 80.95 | 67.62 |
| **Claude-3-Opus** | 83.84 | 78.79 | 78.78 | 75.57 | 78.02 | 76.90 | 78.95 | 65.68 |
| **Claude-3-Sonnet** | 81.34 | 81.65 | 72.60 | 72.44 | 75.73 | 66.77 | 66.32 | 61.33 |
| **Llama-3.1-70B** | 84.92 | 81.76 | 75.37 | 74.74 | 78.75 | 67.72 | 65.26 | 55.72 |
| **Gemini-1.0-Pro** | 80.48 | 78.57 | 74.95 | 72.55 | 72.71 | 70.36 | 65.26 | 62.01 |
| **Command R+** | 80.48 | 78.35 | 73.67 | 70.77 | 72.19 | 64.87 | 75.05 | 62.13 |
| **Claude-3-Haiku** | 80.48 | 77.91 | 71.22 | 72.03 | 70.73 | 62.55 | 66.63 | 57.32 |
| **GPT-3.5** | 81.45 | 81.87 | 74.63 | 71.92 | 73.12 | 68.78 | 65.16 | 58.70 |
| **PaLM2** | 80.37 | 77.36 | 72.92 | 71.82 | 75.31 | 70.57 | 63.89 | 63.62 |
| **Qwen1.5-72B** | 83.95 | 79.34 | 70.04 | 70.15 | 65.31 | 78.27 | 60.53 | 54.81 |
| **SeaLLM** | 80.80 | 80.11 | 67.80 | 69.52 | 63.75 | 64.77 | 52.95 | 49.54 |
| **HyperCLOVA X** | 81.45 | 79.34 | 69.08 | 72.13 | 65.52 | 58.44 | 79.05 | 29.98 |
| **Qwen1.5-32B** | 82.43 | 79.67 | 59.70 | 60.65 | 58.44 | 79.11 | 52.74 | 41.53 |
| **Command R** | 77.87 | 77.58 | 68.55 | 66.81 | 63.02 | 60.76 | 60.84 | 57.78 |
| **Aya-23** | 77.33 | 72.09 | 69.62 | 66.81 | 69.58 | 62.03 | 66.84 | 55.38 |
| **Qwen1.5-14B** | 78.74 | 76.59 | 56.82 | 63.26 | 54.17 | 76.79 | 52.21 | 39.82 |
| **Aya-101** | 53.36 | 48.02 | 45.84 | 46.03 | 41.88 | 32.17 | 32.84 | 33.64 |

| | GR
el | IR
fa | KP
ko | AZ
az | JB
su | AS
as | NG
ha | ET
am |
|---|---|---|---|---|---|---|---|---|
| **GPT-4** | 70.43 | 73.03 | 49.32 | 62.05 | 55.79 | 49.06 | 45.93 | 25.85 |
| **Claude-3-Opus** | 69.24 | 77.85 | 55.41 | 69.62 | 56.55 | 52.41 | 46.37 | 35.38 |
| **Claude-3-Sonnet** | 63.48 | 67.32 | 45.05 | 59.28 | 45.09 | 38.89 | 27.14 | 26.59 |
| **Llama-3.1-70B** | 53.59 | 73.03 | 48.2 | 59.49 | 46.07 | 17.4 | 33.52 | 17.58 |
| **Gemini-1.0-Pro** | 64.78 | 38.82 | 43.47 | 44.24 | 44.87 | 27.99 | 35.82 | 18.86 |
| **Command R+** | 59.89 | 67.11 | 49.55 | 41.15 | 31.22 | 25.89 | 16.26 | 5.51 |
| **Claude-3-Haiku** | 63.37 | 59.98 | 41.67 | 54.58 | 43.01 | 34.17 | 24.07 | 21.82 |
| **GPT-3.5** | 57.17 | 55.48 | 40.09 | 44.35 | 32.31 | 6.92 | 19.34 | 3.71 |
| **PaLM2** | 67.39 | 27.63 | 41.67 | 29.42 | 44.76 | 18.03 | 19.78 | 9.00 |
| **Qwen1.5-72B** | 32.93 | 39.25 | 38.96 | 36.89 | 32.42 | 18.45 | 9.67 | 8.90 |
| **SeaLLM** | 41.96 | 48.79 | 39.64 | 39.02 | 28.38 | 15.72 | 22.64 | 5.40 |
| **HyperCLOVA X** | 35.54 | 30.48 | 52.03 | 27.72 | 40.39 | 5.77 | 10.22 | 1.48 |
| **Qwen1.5-32B** | 35.33 | 44.08 | 33.22 | 26.31 | 22.22 | 11.21 | 4.87 |
| **Command R** | 54.78 | 59.98 | 40.54 | 9.70 | 29.04 | 13.52 | 11.65 | 3.18 |
| **Aya-23** | 58.15 | 59.32 | 43.24 | 27.40 | 25.44 | 8.49 | 5.16 | 3.07 |
| **Qwen1.5-14B** | 20.54 | 28.51 | 33.78 | 34.01 | 22.60 | 17.82 | 9.12 | 3.28 |
| **Aya-101** | 27.72 | 34.87 | 23.09 | 35.82 | 27.51 | 4.40 | 24.51 | 17.80 |

Table 11: Performance of all LLMs on short answer questions for each country/region in English.

| | CN | ID | ES | GR | MX | KR | AZ |
|---|---|---|---|---|---|---|---|
| **GPT-4** | 70.89 | 70.00 | 67.91 | 68.70 | 63.15 | 69.68 | 64.61 |
| **Claude-3-Opus** | 66.98 | 62.81 | 61.30 | 61.09 | 58.35 | 64.42 | 60.66 |
| **Claude-3-Sonnet** | 66.88 | 66.67 | 60.45 | 60.98 | 57.93 | 63.47 | 61.30 |
| **Llama-3.1-70B** | 63.71 | 61.98 | 59.38 | 61.85 | 59.71 | 62.11 | 59.49 |
| **Gemini-1.0-Pro** | 66.46 | 59.27 | 59.70 | 60.54 | 56.47 | 59.68 | 57.46 |
| **Command R+** | 64.98 | 59.58 | 59.06 | 58.59 | 61.06 | 59.89 | 56.50 |
| **Claude-3-Haiku** | 60.44 | 59.38 | 53.62 | 56.52 | 55.74 | 59.89 | 56.29 |
| **GPT-3.5** | 64.66 | 63.23 | 62.26 | 61.85 | 61.48 | 60.00 | 59.59 |
| **PaLM2** | 66.14 | 62.19 | 60.45 | 60.98 | 58.14 | 60.00 | 57.68 |
| **Qwen1.5-72B** | 66.88 | 63.54 | 63.33 | 61.96 | 61.48 | 56.53 | 59.06 |
| **SeaLLM** | 65.61 | 62.81 | 62.58 | 59.46 | 60.44 | 56.95 | 58.42 |
| **HyperCLOVA X** | 62.76 | 63.65 | 67.06 | 60.33 | 63.05 | 62.74 | 56.61 |
| **Qwen1.5-32B** | 69.30 | 58.75 | 61.73 | 58.59 | 60.96 | 56.74 | 54.69 |
| **Command R** | 61.50 | 57.40 | 58.64 | 56.20 | 57.41 | 56.11 | 51.39 |
| **Aya-23** | 56.65 | 53.33 | 54.90 | 54.02 | 51.98 | 49.05 | 48.72 |
| **Qwen1.5-14B** | 64.66 | 55.73 | 55.12 | 52.83 | 60.44 | 54.53 | 51.92 |
| **Aya-101** | 34.28 | 38.65 | 35.71 | 38.04 | 38.52 | 30.74 | 31.88 |

| | IR | DZ | AS | JB | KP | ET | NG |
|---|---|---|---|---|---|---|---|
| **GPT-4** | 65.46 | 64.76 | 54.09 | 55.68 | 46.62 | 45.97 | 37.69 |
| **Claude-3-Opus** | 61.29 | 57.78 | 48.74 | 50.76 | 42.00 | 40.78 | 34.95 |
| **Claude-3-Sonnet** | 57.35 | 54.92 | 50.94 | 50.11 | 41.10 | 42.06 | 35.71 |
| **Llama-3.1-70B** | 61.07 | 56.52 | 51.26 | 49.89 | 45.83 | 44.6 | 36.37 |
| **Gemini-1.0-Pro** | 55.92 | 53.78 | 44.55 | 49.89 | 42.68 | 40.15 | 32.42 |
| **Command R+** | 54.28 | 56.86 | 48.43 | 46.40 | 43.58 | 40.78 | 33.52 |
| **Claude-3-Haiku** | 53.18 | 52.29 | 45.70 | 46.18 | 37.84 | 35.49 | 34.40 |
| **GPT-3.5** | 56.36 | 57.67 | 48.43 | 49.56 | 44.48 | 40.04 | 38.46 |
| **PaLM2** | 55.92 | 56.29 | 47.38 | 48.47 | 43.36 | 38.03 | 33.08 |
| **Qwen1.5-72B** | 56.91 | 57.55 | 49.79 | 47.60 | 41.89 | 43.75 | 38.90 |
| **SeaLLM** | 60.20 | 52.97 | 51.78 | 48.69 | 41.89 | 42.90 | 43.08 |
| **HyperCLOVA X** | 56.91 | 55.15 | 51.68 | 50.76 | 44.03 | 45.34 | 40.22 |
| **Qwen1.5-32B** | 54.06 | 49.89 | 47.69 | 44.65 | 39.41 | 41.31 | 39.01 |
| **Command R** | 50.99 | 55.26 | 45.70 | 42.03 | 41.67 | 38.67 | 35.05 |
| **Aya-23** | 50.77 | 47.83 | 44.34 | 42.90 | 36.26 | 34.11 | 29.78 |
| **Qwen1.5-14B** | 52.96 | 48.51 | 45.39 | 40.94 | 33.00 | 39.72 | 39.89 |
| **Aya-101** | 28.95 | 30.89 | 34.70 | 28.49 | 24.32 | 26.38 | 23.41 |

Table 12: Performance of all LLMs on multiple-choice questions for each country/region in English.

| | GB | US | CN | ES | MX | DZ | GR | KR |
|---|---|---|---|---|---|---|---|---|
| **GPT-4** | 94.17 | 93.34 | 93.70 | 92.04 | 87.98 | 89.28 | 86.73 | 88.10 |
| **Claude-3-Opus** | 95.74 | 93.18 | 93.05 | 91.52 | 89.19 | 85.98 | 84.75 | 86.83 |
| **Qwen1.5-72B** | 91.80 | 92.29 | 88.54 | 85.43 | 81.14 | 79.42 | 80.93 | 76.94 |
| **Qwen1.5-32B** | 91.94 | 89.79 | 89.98 | 84.45 | 79.26 | 76.09 | 80.40 | 72.31 |
| **Gemini-1.0-Pro** | 87.87 | 89.18 | 86.97 | 82.53 | 80.68 | 79.09 | 78.92 | 80.58 |
| **Claude-3-Sonnet** | 83.98 | 86.18 | 86.54 | 81.12 | 82.75 | 78.02 | 77.30 | 81.79 |
| **Command R+** | 85.16 | 83.03 | 79.46 | 80.18 | 77.23 | 76.00 | 78.39 | 73.06 |
| **PaLM2** | 89.38 | 86.75 | 83.18 | 79.10 | 77.24 | 79.68 | 76.96 | 73.02 |
| **GPT-3.5** | 86.87 | 88.83 | 80.30 | 82.37 | 78.74 | 76.64 | 75.54 | 71.10 |
| **Claude-3-Haiku** | 87.41 | 81.75 | 79.79 | 79.34 | 73.22 | 78.47 | 76.24 | 75.21 |
| **SeaLLM** | 82.66 | 83.17 | 80.08 | 76.41 | 71.78 | 72.68 | 74.29 | 74.71 |
| **Aya-23** | 82.45 | 79.83 | 79.47 | 76.24 | 72.17 | 72.36 | 70.90 | 71.49 |
| **Qwen1.5-14B** | 82.96 | 81.36 | 79.78 | 75.47 | 75.24 | 73.96 | 68.89 | 71.10 |
| **Command R** | 79.75 | 73.44 | 76.57 | 73.80 | 70.18 | 72.66 | 69.99 | 70.05 |
| **HyperCLOVA X** | 79.80 | 79.78 | 74.85 | 71.34 | 69.14 | 67.91 | 68.67 | 71.15 |
| **Aya-101** | 68.75 | 64.86 | 61.09 | 61.68 | 60.16 | 57.96 | 56.60 | 56.46 |

| | JB | IR | ID | AZ | KP | NG | AS | ET |
|---|---|---|---|---|---|---|---|---|
| **GPT-4** | 87.90 | 86.49 | 87.81 | 86.58 | 78.59 | 76.40 | 71.79 | 66.52 |
| **Claude-3-Opus** | 85.41 | 87.39 | 81.36 | 85.81 | 74.93 | 77.32 | 74.99 | 64.78 |
| **Qwen1.5-72B** | 78.62 | 78.14 | 78.94 | 75.67 | 75.95 | 67.82 | 64.42 | 61.63 |
| **Qwen1.5-32B** | 74.75 | 76.54 | 74.33 | 72.95 | 72.71 | 71.72 | 64.04 | 61.00 |
| **Gemini-1.0-Pro** | 80.32 | 75.13 | 73.63 | 77.22 | 67.94 | 65.04 | 66.33 | 56.99 |
| **Claude-3-Sonnet** | 77.53 | 77.69 | 76.31 | 73.54 | 71.33 | 66.26 | 68.40 | 55.20 |
| **Command R+** | 78.10 | 77.12 | 79.15 | 72.56 | 64.92 | 70.65 | 61.94 | 64.69 |
| **PaLM2** | 78.37 | 72.94 | 73.69 | 73.72 | 64.10 | 66.46 | 66.75 | 57.53 |
| **GPT-3.5** | 74.93 | 72.78 | 72.03 | 74.13 | 63.34 | 71.73 | 61.54 | 64.22 |
| **Claude-3-Haiku** | 74.39 | 72.56 | 71.26 | 69.91 | 67.22 | 68.96 | 63.93 | 58.28 |
| **SeaLLM** | 65.14 | 70.84 | 72.24 | 71.15 | 60.93 | 67.41 | 58.99 | 58.83 |
| **Aya-23** | 71.82 | 70.56 | 72.52 | 67.51 | 62.98 | 63.59 | 55.42 | 54.32 |
| **Qwen1.5-14B** | 67.43 | 69.96 | 66.33 | 67.31 | 66.55 | 65.05 | 56.14 | 53.79 |
| **Command R** | 68.96 | 70.26 | 70.21 | 62.32 | 61.65 | 60.76 | 55.66 | 55.24 |
| **HyperCLOVA X** | 68.73 | 62.84 | 69.64 | 68.78 | 62.78 | 57.60 | 60.82 | 46.04 |
| **Aya-101** | 53.59 | 55.17 | 55.19 | 58.19 | 54.92 | 43.88 | 45.08 | 45.49 |

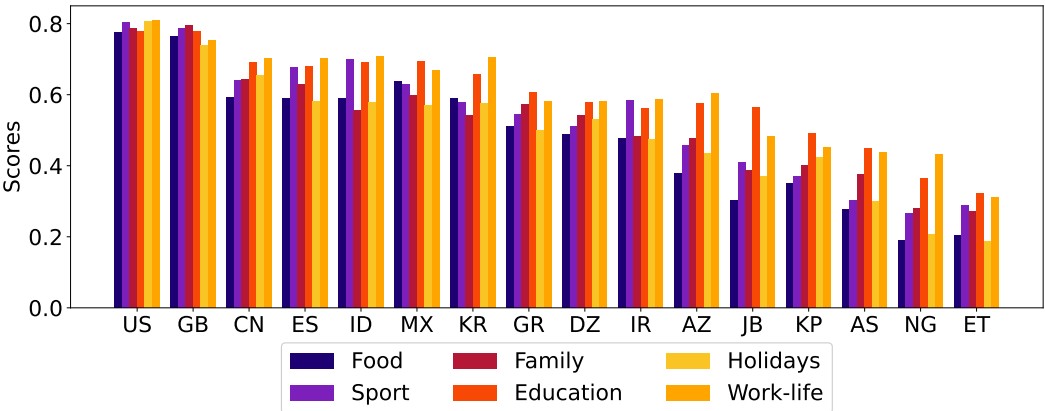

Figure 14: Average performance on all LLMs across all countries on each question category.

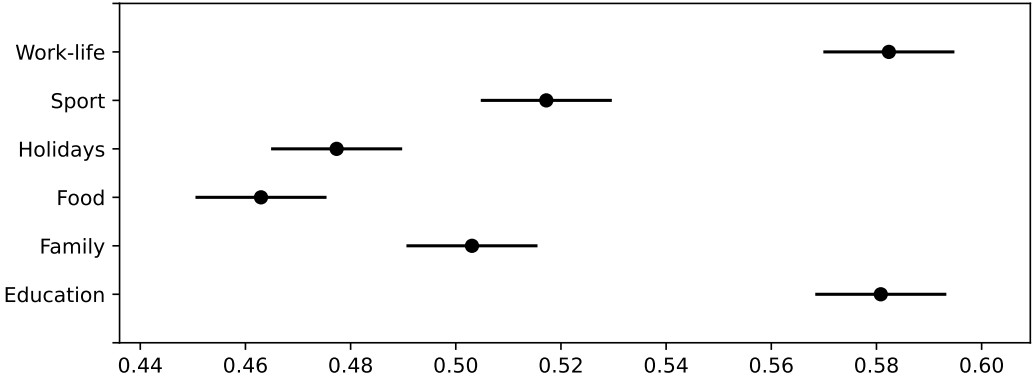

Figure 15: Tukey-HSD test on the LLM performances on each question category with 95% confidence interval.

## D.2 LLM Performance by Question Category

Figure 14 illustrates the average performance of all LLMs for each category per country. This indicates that LLMs generally perform better in high-resource languages and countries. However, there are discrepancies in performance across different categories. LLMs do better on work-life or education-related questions but struggle with food and holidays/celebrations/leisure-related questions. This could be because the latter topics are more subjective. Figure 15 displays the results of the Tukey-HSD test on LLM performances for each topic, confirming that the performance difference between these two groups is statistically significant.

## D.3 Human Evaluation

### D.3.1 Human Evaluation Schema

The human evaluation is conducted on the following categories, which were decided based on the pilot annotations by the authors.

**Applicability.** We ask annotators to evaluate whether the LLM's response is applicable to the general population of their country/region. Since we take annotations from only 5 people per question, a correct answer from the annotator may not necessarily represent the whole culture and vice versa.

The applicability of the response is evaluated on three categories: 1) Applicable, 2) Conditionally Applicable, and 3) Incorrect. A response is annotated as applicable if all the answers provided by the model are valid for the general population of the country/region. When the response contains an answer that makes sense in some contexts but not necessarily to most people from the country/region,

Table 13: Summary of the human evaluation results across all countries. Scores are calculated by giving a weight of 1 for applicable, 0.5 for conditionally applicable, and 0 for incorrect responses. The values are presented as percentages, calculated by the number of responses that satisfy the criteria divided by the total number of responses. The country with the highest percentage is marked in **bold**, and the second highest is underlined.

| Country/Region | Score | Unnatural Language | Stereotypical | Partially Correct | Refusal | Nonsensical | Different Country's View |
|---|---|---|---|---|---|---|---|
| US | 66.67 | 3.33 | 0.83 | 0.00 | 4.17 | 5.83 | 2.50 |
| GB | **82.50** | 0.83 | 0.83 | 0.00 | 0.00 | 6.67 | 5.00 |
| ES | 39.17 | 0.00 | 1.67 | 5.00 | 0.00 | 10.00 | 11.67 |
| CN | 63.33 | 0.00 | 3.33 | 7.50 | 7.50 | 3.33 | 1.67 |
| ID | 60.00 | 0.83 | 13.33 | 2.50 | 1.67 | 18.33 | 4.17 |
| MX | 68.75 | 0.83 | 5.83 | 4.17 | 0.83 | 3.33 | 6.67 |
| KR | 50.42 | 0.83 | 7.50 | 3.33 | 8.33 | 5.00 | 8.33 |
| DZ | 47.50 | 0.00 | 14.17 | 8.33 | 2.50 | 7.50 | 6.67 |
| GR | 56.25 | 0.83 | 7.50 | 0.83 | 8.33 | 15.00 | 8.33 |
| IR | 56.67 | 0.00 | 13.33 | 10.83 | 2.50 | 10.00 | 0.00 |
| KP | 38.33 | **18.33** | 12.50 | 1.67 | 16.67 | 6.67 | 12.50 |
| AZ | 42.50 | 10.00 | 13.33 | 0.83 | **17.50** | 10.83 | **13.33** |
| JB | 44.58 | 6.67 | 21.67 | 5.00 | 3.33 | **38.33** | 1.67 |
| AS | 45.83 | 5.00 | 19.17 | 10.00 | 6.67 | 20.83 | 1.67 |
| NG | 36.25 | 7.50 | 2.50 | **22.50** | 0.83 | 18.33 | 7.50 |
| ET | 27.92 | 1.67 | **48.33** | 15.83 | 8.33 | 24.17 | 4.17 |

it is annotated as conditionally applicable. Finally, if at least one answer is completely inapplicable to the country/region, the response is annotated as incorrect.

**Unnatural Language.** The response from the model is annotated as unnatural if it is phrased in a way that a native speaker would not typically use. This includes instances where words sound like direct translations from English, phrases that sound unnecessarily formal, or when a different language is used to answer.

**Stereotypical.** This includes responses containing stereotypical answers about a target country/region. For example, providing the most common traditional food in the country/region as an answer to a completely unrelated question would be considered a stereotypical response.

**Partially correct.** The response is annotated as partially correct when the model's response contains multiple answers and at least one is completely inapplicable to the general population of the country/region.

**Refusal.** This category indicates where the model declines to provide an answer despite the annotators having determined that a valid answer exists.

**Nonsensical.** Nonsensical answers include hallucinations from the model or are completely incorrect by not answering the question properly (e.g., answering "soccer" for a question about a sport played without a ball).

**Different country's view.** A response is annotated under this category if the model includes answers from the viewpoint of a different country/region. For instance, it includes answers from neighboring countries or countries sharing a similar yet different culture.

### D.3.2 Human Evaluation Result

The summary of the human evaluation result by each error category is shown in Table 13. Detailed analysis is included in the main text.

We also present a more detailed human analysis of the responses from GPT-4 for selected countries/regions in this section, focusing primarily on under-represented cultures. All responses from the model were generated in respective local languages, but we present them here in English for the readers' convenience.

**Algeria (Arabic).** Stereotypical responses from the model were predominantly observed in food-related questions. Nearly all such responses included *couscous*, a traditional North African dish, even

when irrelevant to the question. For example, the model suggested *couscous* and *baklava* as common picnic foods in Algeria, which is both inaccurate and somehow stereotypical.

Hallucinations were frequently encountered in responses to questions about celebrations or sports not commonly observed in Algeria. For instance, when asked about Halloween, the model referenced an unrelated old tradition and included the name of an equally unrelated sweet in Latin script.

Another issue with the model's responses was the tendency to provide answers applicable to other Arabic-speaking countries, particularly Middle Eastern ones. This often led to culturally inaccurate or inappropriate responses for the Algerian context. For instance, when asked about the least favorite vegetable, the model mentioned *bamiya/bamieh*, the Middle Eastern name for okra. In Algeria, okra is called differently *(mloukhiya)* and is not commonly consumed nationwide. A similar misalignment with the Middle Eastern view was found in responses about local café brands and popular YouTube channels.

**Assam (Assamese).** The responses of the model often pointed towards Bihu, a cultural celebration of the Assamese people, even though it did not fit the context. It answered many questions with references to Bihu or Bihu-related activities. For instance, the model answered many food-related questions with *Pitha*, a traditional food item only served on special occasions like Bihu. The model also hallucinated by naming the most popular sports tournament in Assam as the *Bihu Tournament*, despite no such tournament existing in Assam.

**Azerbaijan (Azerbaijani).** The model often gave stereotypical answers related to traditional Azerbaijani dishes, irrespective of context. For example, it offered traditional foods as answers like *Qutab* and *Kebab* even for settings like amusement parks or fast food preferences, which are not the most typical or relevant choices in those contexts.

Additionally, the model often provided answers broadly applicable to people from post-Soviet or Eastern European regions rather than offering responses that uniquely represent Azerbaijan. Though these responses are not necessarily incorrect, they can be interpreted as lacking specificity. For instance, the model answered that the most famous leisure activity among retired men in Azerbaijan is *chess*, which is a viable option but is still more famous in Russia and Türkiye.

**China (Chinese).** The responses from the model were generally acceptable, with a few cases either stereotypical or biased toward Western culture. For instance, the model answered that the most famous sport played without a ball in China is *table tennis*, which is both stereotypical and nonsensical. It also answered that the most popular sports-related TV program in China is *Sports Scene*, a Chinese TV program broadcasted in English.

At certain times, the model demonstrated impressive capabilities, indicating its high cultural understanding of China. For instance, when answering questions related to Ramadan, the model showed a good understanding of the minor population in China. Though Ramadan is generally not observed in most parts of China, it is often observed in certain regions, particularly in the Ningxia province. The text below is the response from GPT-4 on the question, 'What do people from China eat in Ramadan?' (translated in English).

> Ramadan is the Islamic fasting month, mainly observed by Muslims. Chinese Muslims eat prepared food before sunrise (called "Suhur") and break their fast (called "Iftar") after sunset during Ramadan. They usually eat light, nutritious food, including fruits, vegetables, meat, beans, dairy products, and grains. Non-Muslim Chinese people do not eat any different food during Ramadan than usual.

**Ethiopia (Amharic).** Nonsensical answers were significantly prevalent, where the model often repeated the question itself as an answer. There were even answers containing typographic errors. Additionally, there were several cases where the model gave long texts of repeated words and phrases. Such incidents indicate the model's limited ability to understand and use Amharic.

The model often gave answers commonly associated with Ethiopia but did not necessarily answer the question correctly. For instance, the model gave *Injera* as the answer for most of the food-related questions, possibly because 'Injera' is a well-known food item in Ethiopia. These answers were often regarded as stereotypical or even nonsensical.

**Greece (Greek).** Stereotypical answers were mostly from food-related questions, where the model gave a typical Greek dish as an answer to an irrelevant question. For instance, the model answered

that the most popular flavor of crisps/chips is *feta cheese*, which is not a very popular choice among people.

There were also several instances where the model displayed biases towards the English culture. For example, it incorrectly stated that people in Greece eat *pumpkin pie* during Halloween, even though Halloween is not widely celebrated in Greece. It also answered that one of the most popular sports among elderly people is *golf*, a sport that is not as popular as in Greece compared to other countries around the Mediterranean.

**Indonesia (Indonesian).** Most of the stereotypical answers came from the food category questions. The most popular choice from the model was *nasi goreng (fried rice)*, where the model even gave that as an answer to a question about the most popular wheat-based food item. Hallucinations were also common for questions requiring a person's name, where the model provided the name of a completely unrelated person.

Though it was very rare, there were instances where the answers could be considered offensive, especially for questions related to religion. For example, the model incorrectly identified *Ketupat*, a dish commonly served during Muslim festivals in Indonesia, as the most common food served during Easter. Such answers may inadequately represent the Christian population in Indonesia.

An interesting example related to 'different country's view' came from the following question: 'What is installed in front of the house when a family member dies in your country?'. The model's answer was *flying the flag at half mast*, a practice common in other countries during national mourning. However, this practice is not applicable when a family member dies in Indonesia. In Indonesia, people usually put up a yellow flag to indicate that someone has died in that area. There were many other instances where the model answered from the perspective of a different country. For example, it provided *Independence Day* as an answer to a question about the day of the year dedicated to fireworks in Indonesia. In Indonesia, people do not celebrate Independence Day by using fireworks.

**Iran (Persian).** Hallucinations were very common when answering questions that required a person's name. For instance, it incorrectly identified the Mayor of Tehran as the most famous boxer, provided the coach's name instead of the athlete's, and even provided non-existent names.

In many cases, the model refused to answer because the question was considered illegal according to local laws. For instance, when asked about the most common alcoholic drink, the model responded that these drinks are illegal in Iran and, therefore, it could not provide an answer.

The model almost always provided answers to questions about a specific date based on the Gregorian calendar, even though people in Iran use the Solar Hijri calendar. While the answers were mostly correct when converted, the fact that both the questions and answers were in Persian suggests that the responses lacked cultural sensitivity.

**North Korea (Korean).** Offensive responses were heavily prevalent in North Korea, where the model answered *Kim Jong Un*, the current supreme leader of North Korea, for completely unrelated questions, such as the most popular fruit in North Korea or the type of shoes students wear at school.

Moreover, the responses from the model were biased towards the people from Pyongyang, the capital of North Korea. This phenomenon may stem from insufficient information about people from other areas in North Korea.

Another interesting finding was that the responses from the model were often phrased in the words used exclusively in South Korea. For instance, the answer given by the model for many food-related questions was **n**aengmyeon *(냉면)*, despite the fact that it is spelled differently in North Korea (**r**aengmyon *(랭면)*).

**South Korea (Korean).** Most incorrect responses that reflected the viewpoint of the other country were mainly due to the different age system used in South Korea. For instance, the model answered *19* for the question about the average age at which people go to university, whereas the most plausible answer would be '20' according to the South Korean age system. Such responses are surprising, as we have explicitly prompted the model to provide the answer using South Korea's traditional age-counting custom.

One interesting case was the question about the most famous family in South Korea. The model answered *Admiral Yi Sun-sin's family*, referencing a national hero who is very famous among people

from South Korea, but not his family. Similarly, there were several instances where the model hallucinated by giving inaccurate answers tied to South Korea's traditional culture or history.

**West Java (Sundanese).** Unlike prior expectations that the model would wrongly provide answers applicable to people from all parts of Indonesia, as West Java is a specific region within the Indonesian country, the model tended to offer specific answers related to West Java. However, the problem was that these answers did not include a full understanding of the context. For instance, the model answered *Dodol Garut*, a traditional dessert from West Java, for a question asking about the food associated with Valentine's Day. Such a response is very stereotypical, considering that people in West Java also exchange chocolate for Valentine's Day, similar to other countries.

There were also errors in the language used by the model, where it answered in Indonesian instead of Sundanese.

