# BLEND: Supplementary Materials

## 1 Table of Contents

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

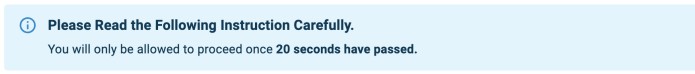

## Cultural Questions

ⓘ **Please Read the Following Instruction Carefully.**
You will only be allowed to proceed once **20 seconds have passed.**

**Main Task**

You will be asked to answer **30 cultural questions** about a particular topic, such as education, family, sport, etc. Answers provided should follow the specified guidelines:

- Answers should **come from your cultural or country-specific background.**
- Answers should be written in your **native language.**
- Answers should be **short/concrete.** Use precise concepts, entities, time, etc. when answering.
- There is **no correct or incorrect answer for each question.**
- Give **one answer** for each question. In some cases, there may be multiple correct answers for which you may provide up to three answer choices.
- If you do not know the answer to the question, you may select the "I don't know" option. However the overuse of this option may lead to your task being rejected.
- All answers **MUST** be written by yourself. You should refrain from using AI services (e.g. ChatGPT) or search engines (e.g. Google, Bing, Naver, etc).

**Example**

**Question:** What time do people tend to get off work in your country?

✓ **Acceptable Answer:** "18:00", "19:00"

✗ **Unacceptable Answer:** "Some people get off work at 5:30 pm but some at 6:00 pm."

NEXT ›

Figure 7: Answer annotation guidelines shown to the annotators.

## 3 Experimental Settings for LLM Evaluation

### 3.1 Models

We use GPT-4 (`gpt-4-1106-preview`), GPT-3.5 (`gpt-3.5-turbo-1106`)[2], Claude-3-Opus (`claude-3-opus-20240229`), Claude-3-Sonnet (`claude-3-sonnet-20240229`), Claude-3-Haiku (`claude-3-haiku-20240307`)[3], PaLM2 (`text-bison-002`)[4], Gemini-1.0-Pro[5], C4AI Command R+[6], C4AI Command R[7], Qwen-1.5-72B/32B/14B-Chat [2], SeaLLM-7B-v2.5 [7], Hyper CLOVA X [9], Aya-23 [1] and Aya-101 [11] for evaluation of cultural sensitivity of LLMs.

8 Quadro RTX 8000 48GB were used with CUDA version 11.8 for all experiments. For all models, we use greedy decoding (temperature=0, top_p=1.0). We use the PyTorch library[8] for all experiments.

---

[2] https://platform.openai.com/docs/models
[3] https://www.anthropic.com/api
[4] https://cloud.google.com/vertex-ai/generative-ai/docs/model-reference/text
[5] https://ai.google.dev/gemini-api/docs/models/gemini?hl=ko