# OpenReview forum: "BLEnD: A Benchmark for LLMs on Everyday Knowledge in Diverse Cultures and Languages"
_NeurIPS.cc/2024/Datasets_and_Benchmarks_Track — NeurIPS 2024 Track Datasets and Benchmarks Poster_

### Official Review · Reviewer_17UX · 2024-07-18
**Review for a hand-crafted cultural benchmark**

**Rating:** 7
**Confidence:** 4
**Correctness:** Yes, they are.
**Clarity:** Yes

**Review:**

The paper introduces a hand-crafted cultural benchmark across diverse cultures and languages, which is valuable. And the construction process is reasonable. There are some points to be improved: (1) More experiments on open-sourced models, such as Llama. (2) More experiments on cultural specific models. (3) More analysis on the results. For example, what may be the factors that influence the cultural understanding ability of LLMs?

Since the number of annotators is really small, the benchmark may also include bias on different cultures.

**Strengths:**

1. The paper introduces a hand-crafted cultural benchmark across diverse cultures and languages, which is valuable.
2. The construction process is reasonable.
3. The paper provides some interesting conclusions on the cultural understanding of LLMs.

**Additional Feedback:**

See above.

**Documentation:**

Yes

**Ethics:**

The number of annotators is really small. The benchmark may also include bias on different cultures.

**Limitations:**

The benchmark may also include bias on different cultures.

**Opportunities For Improvement:**

1. More experiments on open-sourced models, such as Llama.
2. More experiments on cultural specific models.
- MaLA-500: Massive Language Adaptation of Large Language Models
- TAIWAN-LLM: Bridging the Linguistic Divide with a Culturally Aligned Language Model
- Typhoon: Thai Large Language Models
- CulturePark: Boosting Cross-cultural Understanding in Large Language Models
3. What may be the factors that influence the cultural understanding ability of LLMs? The scale of model, analysis on the training corpus. More exploration is more interesting.

**Relation To Prior Work:**

The paper introduces a hand-crafted cultural benchmark across diverse cultures and languages, which is valuable.

**Summary And Contributions:**

This paper introduces a hand-crafted benchmark to evaluate LLM's everyday knowledge across diverse cultures and languages.It also provides two types of questions to automatically measure the cultural knowledge in LLMs: short-answer questions and multiple-choice questions. Furthermore, the authors conduct extensive experiments across 16 LLMs on BLEND, showing a significant 65 performance gap between highly represented cultures and underrepresented cultures.

---

> ### Author Rebuttal · Authors · 2024-08-17
>
> We appreciate all your thoughtful comments on our paper, including the recommendations for open-source and cultural-specific models. We will include the models in the final version of our paper.
>
> Regarding the third point of the factors that influence the cultural understanding ability of LLMs, we believe the training data may have the highest influence on the cultural understanding of LLMs. Such assumptions are drawn from further human analysis, included in Section 5 of the main context. The models often repeated the same answer, for example, the most famous food within the target country, for unrelated questions due to the limited knowledge of diverse cultural entities for underrepresented cultures.
>
> To prevent potential biases from being included in our dataset, we have first explicitly asked the annotators to avoid stereotypical questions, as discussed in lines 111–113. The authors also conducted a manual filtering of the collected questions to remove any questions that could lead to stereotypical answers.

---

> > ### Comment · Reviewer_17UX · 2024-08-28
> > **Reviewer's Response**
> >
> > Thanks for your rebuttal!

---

### Official Review · Reviewer_QRj4 · 2024-07-18

**Rating:** 7
**Confidence:** 3
**Correctness:** YES
**Clarity:** YES

**Review:**

I find this work tackles the important topic of benchmarking everyday cultural knowledge and does so in a principled manner.

**Strengths:**

This paper tackles the important open-challenge of benchmarking how well language models understand every day knowledge for diverse cultures. This is evermore important as language models are widely deployed/used in many applications across cultures.

The authors capture a diverse set of languages and questions across a broad set of topics. I also appreciate that the authors produced question-answer pairs in both English and the native language as well as both short-form and multiple choice questions.

The findings around whether models performed better in Native or English depending on whether the culture used a low or high-resource language is quite interesting.  I also found the qualitative analysis of incorrect responses insightful.

The authors provide a nice repo with questions organized in a clear json format as well as code to reproduce evaluations

**Additional Feedback:**

N/A

**Documentation:**

YES

**Limitations:**

- is my understanding correct that questions differ for each culture (or is the set of questions fixed)? If the questions differ, how did you control for the intrinsic difficulty in questions generated from a particular region? Ensuring 1-2 humans from each region could answer the question is a good filter, but I didn't see a control for difficulty across questions.
- I found the methodology for verifying the correctness of LLM responses for the short-form questions a bit unclear. Checking the presence of the model's response in the human provided response, did not seem like sufficient enough criteria to mark a response as correct (and of course misses out many other potentially correct responses). Can the authors describe this further?

**Opportunities For Improvement:**

See Limitations

**Relation To Prior Work:**

YES

**Summary And Contributions:**

The authors propose a benchmark to evaluate language models' cultural knowledge of everyday life across categories such as food, sports, family, education, holidays, and work-life. The authors release a dataset of ~50k question answer pairs with both multiple choice and short answers across 16 regions.

The authors evaluate 16 large language models, including recently released models such as Gemini, Claude, and GPT-4 both with questions in English and in the native language. The authors find model exhibit a large gap in performance across countries. The authors also analyze a subset of incorrect responses with human annotations revealing incorrect model responses often rely on stereotypes without understanding the questions. The authors also find language models perform better in the native language for higher represented culture and better in English for under-represented cultures.

---

> ### Author Rebuttal · Authors · 2024-08-17
>
> We appreciate your feedback and concerns about controlling the difficulty of the questions across different countries and regions. For the short-answer questions, we employed a fixed set of questions across the 16 countries and regions. To analyze the effect of the implicit difficulty of the questions, we have carried out an additional analysis on the average number of “I don’t know” responses across all countries. Results suggest that the average of such responses was 1.01 out of 5 per question, with a standard deviation of 0.35 (ranging from a maximum of 1.912 in Northern Nigeria to a minimum of 0.42 in South Korea). This suggests that while there were some outliers, the overall difficulty level of the questions was relatively consistent. We will include this information in the appendix to provide further clarity.
>
> Regarding the evaluation of LLM responses for short-form questions, we assessed correctness by determining whether the LLM's response included any of the answers provided by the annotators. This approach is similar to the recall score, which, according to Adlakha et al. [1], shows the highest correlation with human evaluation among all lexical metrics. Details of this evaluation are provided in Section C.2 of the appendix. We will revise lines 168–170 in the main text to clarify this process if you believe that is needed. Regarding the evaluation method’s limitations, as it may not capture all potentially correct or incorrect responses, we have briefly discussed this in lines 287–290. We do acknowledge that our “dataset does not fully represent all the speakers of any language/region” due to the limited number of annotators. To address this issue, we conducted a further human evaluation in our study to assess its alignment with the model performance on our dataset. The scores of the human assessment included in Appendix D.3 show a strong positive Pearson correlation (r = 0.778) with the model’s performance in SAQ using local languages. This indicates that despite the limited number of human annotations, our dataset sufficiently serves as an initial benchmark in evaluating the LLMs’ everyday life knowledge in diverse countries/regions.
>
> [1] Adlakha, Vaibhav, et al. "Evaluating Correctness and Faithfulness of Instruction-Following Models for Question Answering." Transactions of the Association for Computational Linguistics 12 (2024): 775-793.

---

### Official Review · Reviewer_CEVb · 2024-07-23
**LLM Evaluation Benchmark on Diverse Cultures and Languages**

**Rating:** 4
**Confidence:** 4
**Correctness:** Yes
**Clarity:** Yes

**Review:**

Quality: The dataset is available in the code repository alongside necessary scripts for reproducing evaluation results. The scope of the benchmark is impressive and larger than previous works. The Q&A dataset curation involves considerable human annotation effort, which is also commendable.

Clarity: No issues with understanding the paper and its idea. It’s straightforward.

Originality: The idea derives from recent studies of LLMs on non-English languages. Capturing cultural knowledge from online sources is also employed before.

Significance: The work could expect impact in LLM evaluation. Nonetheless, its significance in machine learning is unclear.

**Strengths:**

1. Benchmark potentially has widespread usage in LLM evaluation, as it points to a major weakness of existing LLMs.
2. Major effort, especially human annotations, is required to build this benchmark.
3. This research has a positive social impact because it directs the attention of AI development to less-represented cultures.

**Additional Feedback:**

N/A

**Documentation:**

Available in the provided repository

**Limitations:**

There is a limitations section discussing the lack of annotators and the native speakers for some languages/countries. Unfortunately, I do not consider the technical contribution of the paper enough for acceptance at NeurIPS despite its potential utility for LLM evaluation. The dataset component of the work is satisfactory, but it falls short as a benchmark with only simple evaluation and few insights.

**Opportunities For Improvement:**

1. The benchmark is severely lacking in the empirical experiment section. The results only include a list of accuracy metrics for LLM X dataset pairs, while the analysis is simply an interpretation of the differences in these numbers. One could expect more in-depth data analysis here. Additionally, there are no ablation studies to complement the main results.
2. It is not explained why the included languages and countries are selected. Some justification would make the benchmark more convincing. There is a lack of coverage for South America and Oceania.
3. For some similar languages, it would be interesting to see the correlation in performance and/or performance on a mixture of questions. This includes North Korea/South Korea, Mexico/Spain, Indonesia/West Java, US/GB, Iran/Azerbaijan.
4. Relatedly, how does fine-tuning on one language affect accuracy on a similar language or a more-unrelated language?

**Relation To Prior Work:**

Yes

**Summary And Contributions:**

The benchmark assembles questions and answers from 16 countries across 13 languages, and it provides an evaluation of 16 large language models (LLMs) on this curated Q&A dataset. The results reveal a major discrepancy between performance on high-resource languages and low-resource ones.

---

> ### Author Rebuttal · Authors · 2024-08-17
>
> We appreciate your thoughtful comments on the paper, especially on potential opportunities for improvement. While we generally agree with you on the limitations of our work, please do remember that constructing a high-quality multilingual multicultural dataset is very challenging in many aspects. Hence, there has been a lack of such datasets, and so it is especially important for our dataset to be introduced to the community to enable further progress forward. Having said that, we would like to discuss how we have or plan to address your concerns.
>
> **Experiments**: We appreciate your concerns regarding the limited empirical experiments on the dataset and model performances. Our primary focus of this work was on the development and thorough exploration of the dataset itself. To this end, we conducted several quantitative and qualitative analyses, including evaluating the average level of agreement among annotators—measured by the average of the maximum votes per question—and examining the overlap of answers across countries and regions by analyzing the shared lemmas in the English versions of annotations (lines 145–159). While our work centered on constructing the benchmark and demonstrating its utility in model evaluation, we did include some statistical analyses of model performances. We also compared model performance across different countries and regions, supplemented by human evaluation, to further validate our dataset. We acknowledge that more extensive empirical experiments, including ablation studies, would enhance the robustness of our findings. We are open to your suggestions on improving our work in this area, particularly regarding the types of ablation studies that would be most valuable based on our dataset.
>
> **Language Selection**: We outline the selection process for the chosen languages in lines 103–107. The languages were selected based on varying levels of resource availability as well as geographical and language variations, and we ensured that at least one of our authors was a native speaker of each language. Further details can be found in Table 6 in the Appendix. We acknowledge that our dataset does not encompass countries from every continent. However, we made a considerable effort to ensure regional diversity by including 16 different countries and regions, including underrepresented areas such as Azerbaijan, Ethiopia, and North Korea. In particular, regarding the concern about Latin America, we agree this is a major limitation. Please do note that this dataset construction requires a lot of resources, so the very next step for this research is to include more languages by collaborating with native speakers from additional countries. We believe that presenting this paper as part of the NeurIPS D&B track will accelerate the global collaboration to expand our dataset.
>
> **Comparison of Similar Languages/Cultures**: We appreciate your suggestion to compare the model performance for countries that share the same language. We have made an effort in this direction, as discussed in lines 184–189, where we compared model performance across such countries. Additionally, in the human evaluation section (lines 261–267), we noted that many responses for North Korea appeared to reflect a South Korean perspective, often using language specific to South Korea. However, we did not observe similar trends for Mexico/Spain or UK/US, likely because the cultures of these countries are more prominently featured in global media and are already well-known to the language models. We agree that incorporating more underrepresented countries that speak the same language would be valuable for future work, as it could provide deeper insights into language-specific variations and model performance across different cultural contexts.
>
> Lastly, thank you for the comment on the impact of fine-tuning a model in one culture on its performance in others. For this work, however, the focus was on the development of the benchmark as well as the evaluation of current off-the-shelf models to gain an understanding of the current strengths and limitations of current models for different cultures and languages. We plan to address this issue in our future work.

---

> > ### Author Response · Authors · 2024-08-24
> >
> > Dear reviewer,
> >
> > This is a gentle reminder to kindly review the response we’ve submitted. Your feedback is highly appreciated. Kindly let us know if you have any further queries.
> >
> > Thanks.

---

> > > ### Comment · Reviewer_CEVb · 2024-08-28
> > > **Reply**
> > >
> > > During the revision period, the paper has seen few additions in the empirical section. Reviewer 17UX have suggested some interesting studies to be considered. Therefore, I am maintaining my score.

---

### Official Review · Reviewer_bPDk · 2024-07-24
**BLEnD: A Benchmark for LLMs on Everyday Knowledge in Diverse Cultures and Languages**

**Rating:** 8
**Confidence:** 3
**Correctness:** The claims are correct.
**Clarity:** The paper is well-organized and well-…

**Review:**

### **Quality**
The paper showcases a well-structured approach to addressing cultural and linguistic biases in large language models (LLMs). The methodology is appropriate, and the results are presented with detailed analysis.

### **Clarity**
The paper is clear and well-written. The objectives, methods, and findings are explained comprehensively, with appropriate use of tables and figures to illustrate key points.

### **Originality**
The paper introduces the BLEND benchmark, which fills a gap in the evaluation of LLMs across diverse cultures and languages. The inclusion of low-resource languages and culturally relevant questions adds to the novelty of the research.

### **Significance**
The paper addresses the important issue of cultural and linguistic biases in LLMs. The BLEND benchmark is a useful tool for researchers and developers to improve the cultural adaptiveness of LLMs, promoting more inclusive and fairer AI-based systems.

### **Cons**
The number of annotators per country is relatively small, which may not fully capture the cultural diversity within each country. The focus on short-answer and multiple-choice questions, while useful, might miss out on evaluating long-form responses and interactive dialogue capabilities of LLMs. The human evaluation of errors introduces subjectivity, which could be mitigated by incorporating more automated evaluation methods. The academic background and English proficiency of annotators might introduce bias, as they may not represent the general population of their respective countries. While BLEND covers 16 countries, there are still many cultures and languages that are not included, which could be addressed in future expansions of the benchmark.

**Strengths:**

### **Significance of its Contribution**
The paper's primary contribution, the BLEND benchmark, helps in the evaluation of LLMs. It provides a tool for assessing cultural knowledge and adaptability across diverse cultures and languages, which is critical for developing fair and inclusive AI systems. By including low-resource languages, the paper addresses a crucial gap in AI research. This focus helps in understanding and mitigating the biases that LLMs exhibit towards underrepresented languages and cultures. The paper offers a solid analysis of the performance gaps and biases in LLMs, providing valuable insights into the areas where these models need improvement.

### **Relevance to the Broader Research Community**
The paper highlights the importance of cultural and linguistic diversity in AI training data, encouraging the broader research community to consider these factors in their work. The BLEND benchmark helps in evaluating LLMs and can serve as a possible starting point for future studies that seek to improve the cultural adaptiveness of AI models. T

### **Quality of the Research**
The research employs a methodology that is appropriate for the construction of a benchmark for the evaluation of LLMs. The use of native speakers and culturally relevant questions helps to ensure the authenticity and reliability of the benchmark. The paper provides a thorough evaluation of multiple LLMs using both short-answer and multiple-choice questions, as well as human evaluations for error analysis. This multi-faceted approach enhances the credibility of the findings.

### **Ethical and Social Implications**
The paper underscores the ethical imperative to develop AI systems that are fair and unbiased by identifying and highlighting biases in LLMs. The paper’s findings and recommendations contribute to the broader discourse on responsible AI development. By providing tools and insights to mitigate biases, it supports the creation of systems that are more aligned with ethical standards and societal values.

**Additional Feedback:**

I have no additional feedback.

**Documentation:**

Sufficient detail is provided in the paper and supplemental materials.

**Ethics:**

There are no ethical concerns.

**Limitations:**

The paper articulates the limitations of the benchmark and reasonably identifies what needs to be done to address the shortcomings.

**Opportunities For Improvement:**

The shortcomings of the paper have already been addressed. There are minor improvements that can be made, but it would not increase the quality of the paper.

**Relation To Prior Work:**

The paper's contribution is clear.

**Summary And Contributions:**

The paper addresses the gap in evaluating large language models (LLMs) for cultural sensitivity and everyday knowledge across diverse regions and languages. Existing benchmarks are limited in scope, often focusing on single languages or highly represented cultures, and failing to capture the nuances of daily life in underrepresented regions. The paper underscores the need for diverse training data to improve LLMs' cultural adaptiveness and reduce biases, providing a valuable resource for researchers and developers working on culturally aware AI systems.  The paper makes the following contributions:

1. **Introduction of BLEND Benchmark** that comprises 52.6k question-answer pairs from 16 countries/regions in 13 languages, including low-resource languages such as Amharic, Assamese, Azerbaijani, Hausa, and Sundanese. It is designed to evaluate LLMs on everyday cultural knowledge with questions in both short-answer and multiple-choice formats.

1. **The study reveals significant performance gaps** between highly represented and underrepresented cultures in LLMs. For instance, GPT-4, the best-performing model, shows a performance difference of up to 57.34% between cultures.

1. **The benchmark demonstrates that LLMs perform better in local languages** for mid-to-high-resource cultures but perform better in English for low-resource cultures.

---

> ### Author Rebuttal · Authors · 2024-08-17
>
> We sincerely appreciate your thoughtful review and feedback. Below, we address the key concerns you raised.
>
> **Number of Annotators**: First, we recognize the limitations associated with the relatively small number of annotators per country and the potential biases this may introduce, as noted in lines 281–283. While we do not claim that our data fully represents all speakers of any language or region, we firmly believe that our dataset offers a valuable foundation for researchers exploring this topic, especially given that the questions and answers were manually crafted by native speakers. We agree that expanding the number of annotators per country, ensuring demographic diversity, and including more languages and regions in the dataset are critical areas for future work.
>
> **Format of Evaluation**: We also acknowledge the limitations of using short-answer and multiple-choice questions, as highlighted in lines 293–296. We fully understand your concerns and are already planning to explore more complex and nuanced cultural questions in future research. This includes evaluating LLMs' capabilities to engage with longer, more contextually rich content, as well as exploring new evaluation methodologies.

---

### Official Review · Reviewer_a2x3 · 2024-07-26
**Valuable benchmarks and extensive analysis**

**Rating:** 7
**Confidence:** 4
**Correctness:** The motivation, data collection, and …
**Clarity:** The paper is well-written, with clear…

**Review:**

Please see below stengths, improvement and limitations.

**Strengths:**

1. The proposed dataset includes a wide range of cultures and languages, and also low-resource ones.

2. Clear explanation of data collection, question filtering, and answer annotation processes.

3. Thorough evaluation of multiple LLMs, showcasing their strengths and weaknesses in cultural knowledge.

4. The topic is interesting and I believe it will be valuable to our community.

**Additional Feedback:**

No

**Documentation:**

Good writing.

**Ethics:**

No ethical concerns.

**Limitations:**

1. Using translation to construct multilingual dataset.

2. The benchmark uses short-answer and multiple-choice questions, which might not fully capture the complexity of cultural knowledge.

**Opportunities For Improvement:**

1. I believe there may be some translation errors when converting questions from English to local languages. How do you address and mitigate potential translation errors in the questions?

2. Some questions, such as "What food is associated with Halloween in your country?" may not be relevant to certain cultures, such as Chinese culture. How do you ensure the meaningfulness of probing LLMs with culturally appropriate questions?

3. The questions and answers in the dataset are designed to be simple, which may facilitate easier understanding and post-processing by LLMs. However, this simplicity could also reduce the dataset's overall value. How do you address this trade-off between simplicity and the richness of cultural knowledge assessment?

**Relation To Prior Work:**

The authors clearly discusses how BLEND differs from previous works.

**Summary And Contributions:**

The paper introduces BLEND, a benchmark designed to evaluate LLMs' cultural knowledge across diverse cultures and languages. The benchmark comprises 52.6k question-answer pairs from 16 countries/regions in 13 different languages, including low-resource languages. It addresses the gap in current benchmarks by focusing on everyday cultural knowledge often missing from online sources like Wikipedia. The study reveals significant performance distinction and challenges among LLMs, particularly in low-resource languages.

---

> ### Author Rebuttal · Authors · 2024-08-17
>
> We appreciate the feedback on the limitations and potential opportunities for enhancing our work. Below, we outline how we have addressed these limitations in our current study and how we plan to tackle them in future research.
>
> **Translation Errors**: We fully recognize the importance of accurate translation in ensuring the integrity of our dataset. To mitigate potential translation errors, we engaged native speakers fluent in both English and their respective local languages to manually translate all questions, as detailed in lines 118–119. Moreover, we continually tracked the responses from the annotators to flag the question when inaccurate answer choices were identified. The native speakers then reviewed the flagged questions to ensure the accuracy and cultural relevance of the translations. We believe these measures substantially reduced the risk of translation errors.
>
> **Cultural Relevance**: We agree that certain questions may not be relevant across all cultural contexts. As mentioned in lines 130–131, we anticipated this issue and included an option for annotators to mark a question as “not applicable to our culture.” Instead of removing such questions, we assume that including such data is valuable for assessing the models' cultural adaptability and their handling of unanswerable questions. This is because LLMs should possess the ability to recognize when a question is unanswerable within a given cultural context rather than generating potentially erroneous or culturally insensitive responses.
>
> **Simplicity vs. Richness**: We acknowledge the concern regarding the simplicity of the questions and answers in the dataset. The straightforward nature of the questions was intentional, aiming to ensure clarity and minimize ambiguity across different languages and cultural contexts, as well as to provide a simpler and fair evaluation. We do emphasize that this is a starting point. Future work will explore more complex and nuanced cultural questions, including those that evaluate LLMs' ability to engage with longer and more contextually rich content. This is also mentioned in lines 293–296 in the limitations section.

---

### Author Response · Authors · 2024-08-17

We authors sincerely thank each reviewer for the time and effort invested in reviewing our paper. The constructive feedback will enhance both the quality and clarity of our work. We are committed to addressing each point raised in your reviews and will outline below how we plan to revise our paper based on your valuable feedback.

---

> ### Author Response · Authors · 2024-08-28
>
> Dear Reviewers,
>
> This is a friendly reminder that the author-reviewer interaction period will end on August 31st. We would greatly appreciate your feedback on the rebuttal.
>
> Thank you once again for your time and dedication to the reviewing process.

---

### Decision · Program_Chairs · 2024-09-26

**Decision:**

Accept (Poster)

**Comment:**

The paper introduces BLEND, a benchmark for evaluating large language models (LLMs) on cultural knowledge across diverse cultures and languages, including low-resource ones. Comprising 52.6k question-answer pairs from 16 countries in 13 languages, it addresses a gap in existing benchmarks by focusing on everyday cultural knowledge. Reviewers highlighted the benchmark’s broad cultural and linguistic scope, clear data collection methods, and thorough evaluation of LLMs, noting its value in promoting culturally inclusive AI development (Reviewer 1, 2, 4, 5). The benchmark was praised as a significant and impactful contribution to the field, providing a useful resource for researchers and developers (Reviewer 2, 3).

Reviewers identified potential translation errors and cultural irrelevance of some questions, which may affect the dataset’s validity (Reviewer 1). The simplicity of questions was seen as a trade-off that could reduce the benchmark’s richness (Reviewer 1). Concerns were raised about the small number of annotators per country, the limited depth of empirical analysis, and the absence of ablation studies (Reviewer 2, 3, 5). The selection of languages and countries was not well-justified, and the focus on short-answer and multiple-choice questions limited the evaluation of LLMs' capabilities in more complex contexts (Reviewer 2, 3). Additionally, the human evaluation of errors introduced subjectivity, and the benchmark's cultural and linguistic coverage was noted as incomplete (Reviewer 2, 3, 4).

Overall, since most of the reviewers gave strong ratings to the paper and it has no significant drawbacks that cannot be solved, this paper should be accepted.